# MR$^2$-Bench: Going Beyond Matching to Reasoning in Multimodal Retrieval

## Abstract

Multimodal retrieval is becoming a crucial component of modern AI applications, yet its evaluation lags behind the demands of more realistic and challenging scenarios. Existing benchmarks primarily probe surface-level semantic correspondence (e.g., object–text matching) while failing to assess the deeper reasoning required to capture complex relationships between visual and textual information. To address this gap, we introduce MR$^2$-Bench, a reasoning-intensive benchmark for multimodal retrieval. MR$^2$-Bench presents the following critical values: 1) all tasks are reasoning-driven, going beyond shallow matching to effectively assess models' capacity for logical, spatial, and causal inference; 2) it features diverse multimodal data, such as natural images, diagrams, and visual puzzles, enabling comprehensive evaluation across content types; 3) it supports complex queries and documents containing multiple images and covers diverse retrieval scenarios, more accurately reflecting real-world applications. Our benchmark contains 1,309 curated queries, derived either from manual collection and annotation or from selective consolidation of public datasets. Despite achieving strong results on existing benchmarks, current state-of-the-art models still struggle on MR$^2$-Bench: for example, the leading Seed1.6-Embedding model attains a Recall@1 of 77.78 on MMEB, but only 9.91 on MR$^2$-Bench. This substantial performance gap highlights both the increased challenge posed by our benchmark and the pressing need for further advances in reasoning-intensive multimodal retrieval.

## 1 Introduction

Multimodal retrieval is a crucial capability in contemporary AI applications, supporting tasks such as image search (Young et al., 2014; Zhang et al., 2024), retrieval-augmented generation (RAG) (Chen et al., 2022; Yu et al., 2024), and multimodal agentic systems (Geng et al., 2025; Wu et al., 2025). The field has evolved from traditional cross-modal matching (e.g., text-to-image retrieval (Chen et al., 2015)) to more advanced multimodal retrieval that accommodates compositional queries over interleaved image-text content (e.g., composed image retrieval (Baldrati et al., 2023) and multimodal knowledge retrieval (Chang et al., 2022; Luo et al., 2023)). Consequently, modern multimodal retrievers (Zhou et al., 2024; Zhang et al., 2025a; Meng et al., 2025) can process queries expressed in text, images, or combinations thereof, efficiently extracting relevant information from diverse data sources and bridging the gap between complex datasets and real-world user needs.

Despite these advances, current evaluation methods remain misaligned with practical requirements. First, existing benchmarks primarily assess surface-level semantic correspondence, offering limited coverage of knowledge reasoning, spatial perception, and vision-centric challenges critical for diverse agentic applications. Second, these benchmarks predominantly feature natural images, with insufficient representation of visual puzzles, diagrams, and mathematical figures common in technical and educational contexts. Third, real-world documents often exhibit free-form, interleaved image-text layouts with multiple images positioned arbitrarily within the text. However, current benchmarks frequently limit each example to a single image (Chang et al., 2022; Baldrati et al., 2023; Hu et al., 2023; Jiang et al., 2025), failing to reflect the complex document structures prevalent in practice. These limitations hinder rigorous evaluation of multimodal retrieval systems in reasoning-intensive, real-world scenarios.

| Benchmarks | #Queries | #Tasks | Multi-Modality | Reasoning-Intensive | Vision-Centric Reasoning | Multi-Domain | Free-Form |
|---|---|---|---|---|---|---|---|
| MS MARCO (Bajaj et al., 2016) | 5,193 | 1 | ✗ | ✗ | ✗ | ✗ | ✗ |
| BEIR (Muennighoff et al., 2022) | 54,262 | 18 | ✗ | ✗ | ✗ | ✓ | ✗ |
| RAR-b (Xiao et al., 2024a) | 45,745 | 17 | ✗ | ✓ | ✗ | ✓ | ✗ |
| BRIGHT (Hongjin et al., 2025) | 1,384 | 12 | ✗ | ✓ | ✗ | ✓ | ✗ |
| CIRR (Liu et al., 2021) | 4,148 | 1 | ✓ | ✗ | ✗ | ✗ | ✗ |
| WebQA (Chang et al., 2022) | 7,540 | 1 | ✓ | ✗ | ✗ | ✗ | ✗ |
| M-BEIR (Wei et al., 2024) | 190,000 | 10 | ✓ | ✗ | ✗ | ✓ | ✗ |
| ViDoRe (Faysse et al., 2025) | 3,810 | 2 | ✓ | ✗ | ✗ | ✗ | ✗ |
| MMEB (Jiang et al., 2025) | 36,000 | 36 | ✓ | ✗ | ✗ | ✓ | ✗ |
| **MR$^2$-Bench (Ours)** | 1,309 | 12 | ✓ | ✓ | ✓ | ✓ | ✓ |

Table 1: Comparison of MR$^2$-Bench with existing benchmarks. Columns report the number of test queries (**#Queries**); the number of tasks (**#Tasks**); inclusion of image–text data (**Multi-Modality**); whether the benchmark is explicitly reasoning-focused (**Reasoning-Intensive**); whether it contains tasks solvable purely from images without textual cues (**Vision-Centric Reasoning**); domain coverage (**Multi-Domain**); and support for arbitrary text–image organization—interleaved ordering and multi-image on the query and document sides (**Free-Form**). The first block represents textual retrieval benchmarks, and the second block represents multimodal retrieval benchmarks.

In this paper, we introduce **MR$^2$-Bench** (Multimodal Reasoning-intensive Retrieval Benchmark). We summarize the key features of MR$^2$-Bench compared to existing benchmarks in Table 1. In summary, MR$^2$-Bench presents the following critical advantages:

- **It is the first benchmark for multimodal reasoning-intensive retrieval.** MR$^2$-Bench is pioneering in its requirement for reasoning to capture relevance rather than relying on shallow semantic matching, thereby filling a significant gap in current multimodal retrieval benchmarks. While existing text-only reasoning-intensive retrieval benchmarks (Xiao et al., 2024a; Hongjin et al., 2025) have been developed, MR$^2$-Bench emphasizes multimodal capabilities with a variety of visually related reasoning-intensive retrieval tasks.

- **It introduces a broad range of multimodal data domains.** Beyond typical natural images, MR$^2$-Bench incorporates diverse image types such as mathematical visual proofs, visual puzzles, and economic charts, etc. These images have widespread applications and inherently require visual reasoning capabilities. However, previous multimodal retrieval tasks have largely overlooked these data types.

- **It offers diverse evaluation scenarios.** MR$^2$-Bench encompasses three meta-tasks: multimodal knowledge retrieval, visual illustration search, and visual relation reasoning, totaling 12 sub-tasks. These tasks provide a wide array of retrieval scenarios, including text-to-image, image-to-image, and mixed image-text queries, among others. Moreover, unlike previous multimodal benchmarks where queries or documents typically contain at most a single image (Wei et al., 2024; Jiang et al., 2025), both queries and documents in MR$^2$-Bench may include multiple images, more accurately reflecting real-world scenarios.

We conduct comprehensive evaluation experiments on existing methods and derive the following key conclusions. Firstly, *multimodal reasoning-intensive retrieval remains challenging for current retrievers*. Despite Seed1.6-Embedding (Seed, 2025) achieves the best performance on MR$^2$-Bench, it only reaches 30.68 nDCG@10. In contrast, it attains 77.78 Recall@1 on the MMEB dataset (Jiang et al., 2025), while its MR$^2$-Bench Recall@1 is just 9.91. Consistent failures are observed across all methods, particularly in mathematical visual proofs and visual relation reasoning. Secondly, *the capability of visual understanding plays an important role in solving our benchmark*. On the one hand, augmenting text-only retrievers with image captions yields substantial gains compared to ignoring images. On the other hand, despite current multimodal retrievers not being optimized for reasoning-intensive retrieval, the two strongest methods in our evaluation are native multimodal retrievers. Finally, *reasoning capacity holds significant potential for enhancing performance on MR$^2$-Bench*. We implement reasoning-enhanced strategies including query rewriting and reranking, which have demonstrated substantial improvements on MR²-Bench. These insights highlight the challenges and opportunities in multimodal retrieval. By exposing current strengths and weaknesses, we anticipate that MR$^2$-Bench will guide the development of more capable multimodal retrievers.

## 2 RELATED WORK

**Reasoning-intensive Retrieval.** Information retrieval (IR) has advanced from lexical matching (Robertson et al., 2009) to capturing deep semantic relevance (Karpukhin et al., 2020; Xiao et al., 2024b; Zhang et al., 2025b). Recently, the rise of applications like retrieval-augmented generation and agentic systems (Li et al., 2025b; Jin et al., 2025; Qian & Liu, 2025) has spurred the need for a more advanced capability: reasoning-intensive retrieval. This paradigm challenges IR systems to address complex information needs where relevance cannot be determined by direct semantic overlap, but must be inferred through deep reasoning. Although there has been significant progress in text-only domains with pioneering benchmarks such as BRIGHT (Hongjin et al., 2025) and the development of specialized retrievers (Shao et al., 2025; Long et al., 2025), its application to multimodal scenarios remains largely unexplored. Our work addresses this gap for the first time. Beyond knowledge-oriented tasks, we introduce novel, vision-centric challenges, including visual illustration search and visual relational reasoning, requiring models to perform complex inference over integrated visual and textual data.

**Multimodal Retrieval.** As real-world information is increasingly presented in multimodal formats, multimodal retrieval has become essential for effectively searching corpora that integrate text and visual data. Initially, the focus was on cross-modal retrieval, such as text-to-image searches (Chen et al., 2015). The field has since evolved to tackle more complex tasks, including image searches guided by textual instructions (Wu et al., 2021; Zhang et al., 2024), multimodal document retrieval (Chang et al., 2022), and knowledge retrieval using multimodal queries (Luo et al., 2023). With the advent of powerful pre-trained vision-language models (VLMs), researchers have been able to develop unified embedding models that effectively handle queries and documents in various formats (Lin et al., 2024; Zhou et al., 2025). Despite these advances, existing benchmarks and methods have largely concentrated on shallow semantic alignment or instance-level matching, neglecting the complex reasoning required to address many real-world information needs (Wei et al., 2024; Jiang et al., 2025). Moreover, these benchmarks often emphasize natural images, overlooking visually complex and abstract domains that demand visual-centric reasoning abilities, such as visual puzzles, mathematical diagrams, and multi-image relational scenarios. Consequently, there is a pressing need for a benchmark designed to evaluate deeper reasoning capabilities in multimodal retrieval.

## 3 MR$^2$-BENCH: MULTIMODAL REASONING-INTENSIVE RETRIEVAL BENCHMARK

We propose MR$^2$-Bench, the first multimodal reasoning-intensive retrieval benchmark. A brief overview of MR$^2$-Bench's statistics is presented in Table 2, and visual examples for each task type are shown in Figure 1. MR$^2$-Bench comprises 3 meta-tasks and 12 sub-tasks, encompassing a total of 1,309 queries. Detailed modalities of queries and documents, along with the instructions for each sub-task, are provided in Appendix C.

| Meta-task | Multimodal Knowledge Retrieval | | | | | | Visual Illustration Search | | | Visual Relation Reasoning | | | Total |
|---|---|---|---|---|---|---|---|---|---|---|---|---|---|
| Sub-task | Biology | Cooking | Gardening | Physics | Chemistry | EarthScience | Economics | Mathematics | Nature | Spatial | Puzzle | Analogy | – |
| #Queries | 79 | 76 | 129 | 76 | 124 | 99 | 84 | 86 | 100 | 149 | 160 | 147 | 1,309 |
| #Corpus | 4,455 | 2,786 | 5,636 | 6,656 | 4,317 | 3,014 | 7,572 | 944 | 2,017 | 1,000 | 5,375 | 3,970 | 47,742 |

Table 2: Data statistics of queries and corpus for each sub-task in MR$^2$-Bench

### 3.1 MULTIMODAL KNOWLEDGE RETRIEVAL

Traditional knowledge retrieval has focused primarily on text-only queries and corpora (Chen et al., 2017; Kwiatkowski et al., 2019). However, images play a crucial role in realistic knowledge retrieval scenarios. For instance, when users wish to explore an intriguing scientific phenomenon in their daily lives, capturing an image for querying is often more intuitive and detailed than using text alone. Similarly, knowledge bases frequently integrate text and images, with images providing essential explanatory and knowledge representation functions. Although some benchmarks have been developed for multimodal knowledge search (Chang et al., 2022; Luo et al., 2023; Hu et al., 2023; Chen et al., 2023), they are predominantly based on annotations from sources like Wikidata, with

questions that are often straightforward (e.g., *What is this mountain called?*[1]). These tasks typically rely on keyword matching, image instance matching, or simple shallow semantic alignment. However, real-world user queries can be highly complex, requiring intensive reasoning to identify relevant documents.

BRIGHT (Hongjin et al., 2025) introduced the first benchmark for evaluating reasoning-intensive knowledge retrieval by constructing retrieval pairs between real user queries from Stack Exchange[2] and relevant documents. The relevant documents are identified from external links referenced in high-scoring answers, establishing retrieval relationships that require reasoning over critical concepts or theories to bridge the query and the document. As a result, retrieval models evaluated on this benchmark must possess capabilities that go beyond simple lexical or semantic matching. However, BRIGHT is a text-only benchmark, leaving a gap in multimodal queries and documents.

Inspired by BRIGHT's task construction approach, we have developed a set of reasoning-intensive multimodal knowledge retrieval tasks in our MR$^2$-Bench. In contrast to BRIGHT, our approach rigorously ensures that images are essential components of the questions, rendering these inquiries invalid without the accompanying visual data. We also retain images from relevant documents if they are crucial for conveying knowledge. The annotation process is detailed in the Appendix D. Our benchmark covers six domains: **Biology**, **Cooking**, **Gardening**, **Physics**, **Chemistry**, and **Earth Science**. Examples of these tasks are illustrated in Figure 1(a)-(c). For instance, in Figure 1(a), the positive document does not mention *apple* or *grow together*. The key to connecting the document and the question lies in the accompanying image, which demonstrates a similar biological phenomenon in other species.

### 3.2 VISUAL ILLUSTRATION SEARCH

Text-to-image retrieval (e.g., Flickr30K (Young et al., 2014), MSCOCO (Chen et al., 2015)) is a canonical multimodal retrieval task, where the system need to retrieve the image that best matches a textual query. Classic benchmarks are largely limited to direct and surface-level semantic alignment, such as identifying a specific animal or a person performing a certain sport. However, real-world use cases often require domain knowledge and multi-step reasoning to retrieve the target image (e.g., professional charts and scientific illustrations). To address this gap, we introduce the **Visual Illustration Search** (VIS) task. In this task, the model is required to retrieve an image that functions as a visual illustration, intuitively explaining or solving a problem posed in a challenging, domain-specific textual query. Comprising three sub-tasks: **Economics**, **Mathematics**, and **Nature**, VIS evaluates a model's ability to perform cross-modal reasoning and knowledge-grounded understanding in complex multimodal scenarios.

**Economics**. Charts serve as intuitive illustrations across various disciplines. However, existing chart-related tasks (e.g., ChartVQA (Masry et al., 2022), ViDoRe (Faysse et al., 2025)) primarily test surface-level abilities solvable with basic OCR and arithmetic. To assess a model's ability to capture the deeper semantics and domain knowledge embedded in chart, we manually collected reports from the World Bank[3], extracted charts related to economics, and asked human experts to create questions grounded in these charts. The core annotation principle is that each question must demand sufficient reasoning to identify the positive chart. For instance, as shown in Figure 1(d), the positive chart does not explicitly state the conclusion; only by comparing the relative positions of different countries in the chart and associating *spending quantiles* with *learning poverty rates* can one validate the hypothesis posed in the question. Following these principles, we constructed a reasoning-oriented retrieval subset centered on economic charts, comprising 84 high-quality questions.

**Mathematics**. Images can effectively reinforce human comprehension of abstract knowledge. This holds especially in mathematics, where *visual proofs* are conical examples that use geometric relations to demonstrate abstract theorems intuitively. As shown in Figure 1(e), the recursive partition of the unit square gives a clear proof of the infinite series $\sum_{n=1}^{\infty} \frac{1}{2^n} = 1$. Although structurally simple, such proofs embody rigorous logic and require strong reasoning to connect visual patterns with abstract mathematical principles, providing an effective evaluation of model's reasoning ability. However, visual proofs are largely absent from existing multimodal retrieval benchmarks. There-

---

[1]Query example curated from the OVEN benchmark (Hu et al., 2023)

[2]https://stackexchange.com

[3]https://data.worldbank.org

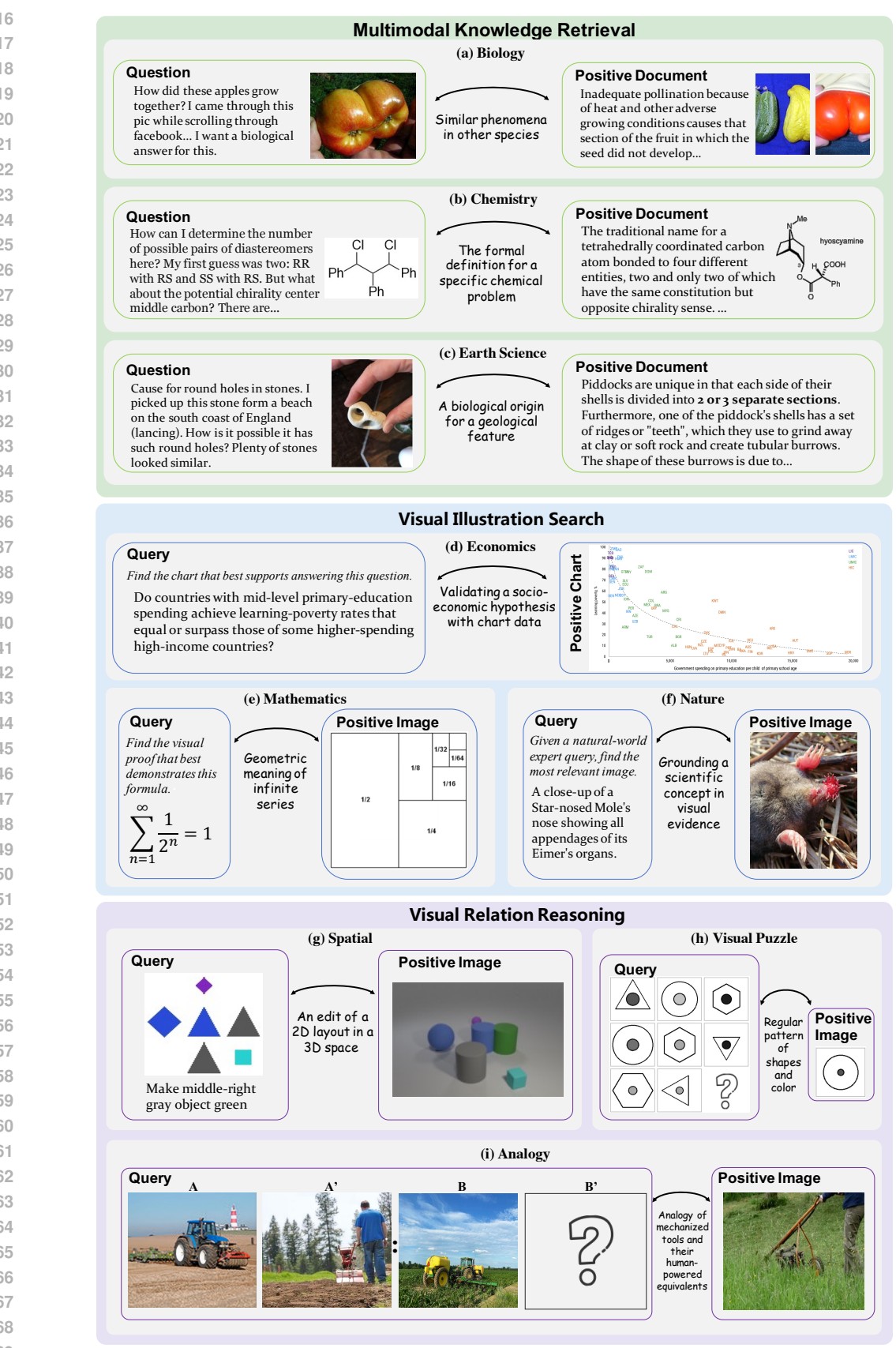

Figure 1: Visualized Examples of MR$^2$-Bench: Sub-task illustrations from three meta-tasks, with 3 out of 6 shown for the multimodal knowledge retrieval task.

fore, we curate 86 mathematical formulas from *Proofs Without Words* (Nelsen, 2015) and Wikimedia Commons[4], using each formula as a query and its corresponding visual proof as the positive image.

**Nature**. Natural-world images are more than depictions; they are visual reference for species identification, ecosystem monitoring, and science education (Van Horn et al., 2015; 2018), which require images that capture specific traits or morphology, rather than the generic picture of the organism. For example, as shown in Figure 1(f), the query seeks for *a close-up of star-nosed mole's distinctive organs*, which demands both expert biological knowledge and fine-grained visual recognition. Satisfying such knowledge-intensive visual requests is a challenging yet essential capability for models. To evaluate this, we carefully selected 100 queries from the publicly available INQUIRE-Rerank dataset (Vendrow et al., 2024) to construct the expert-level natural-world image retrieval task.

### 3.3 VISUAL RELATION REASONING

In prevailing multimodal retrieval benchmarks, textual queries are the primary driver of user intent. However, this paradigm often overlooks the rich, self-contained semantics inherent in purely visual structures and relationships that are independent of natural language. To address this gap, we introduce **Visual Relation Reasoning**, a suite of tasks for assessing high-level vision-centric reasoning through three distinct sub-tasks: **Spatial**, **Visual Puzzle**, and **Analogy**.

**Spatial**. The capacity for spatial perception, transformation, and reasoning is essential for models. To evaluate these capabilities, we incorporate tasks from the CSS dataset (Vo et al., 2019), a controlled synthetic dataset where each sample consists of a reference image, a textual modification instruction, and a corresponding target image, with scenes rendered as both 2D layouts and photorealistic 3D images. As illustrated in Figure 1(g), the query requires jointly parsing descriptions that combine relative position and attributes (i.e., *middle-right gray object*) and projecting the 2D layout into the corresponding 3D scene, yielding a comprehensive test of spatial ability. From CSS, we curated 149 queries to constitute the spatial-reasoning subtask of $MR^2$-Bench.

**Visual Puzzle**. Inspired by Raven's Progressive Matrices[5], this task is designed to evaluate pattern recognition and structural reasoning. As shown in Figure 1(h), for a given 3×3 matrix with the final cell missing, the model need to retrieve the positive image that logically completes the matrix's underlying pattern. This task is distinguished by its near-complete absence of linguistic signals, which compels the model to directly infer abstract patterns to perform higher-order reasoning from vision alone. We reorganized the RAVEN dataset (Zhang et al., 2019): for each rule-governed visual attribute, we selected a set of queries, pooled the corresponding candidate images and removed duplicates to build the corpus. In total, we curated 160 queries for this task.

**Analogy**. Derived from the VASR dataset (Bitton et al., 2023), this task tests a model's capability for visual analogical reasoning. As shown in Figure 1(i), the query comprises three images $(A, A', B)$, where the pair $(A, A')$ exemplifies a visual semantic transformation (e.g., *replacing a machine with human labor in a comparable scene*) that is expected to hold between $B$ and $B'$. The model must infer the transformation from $A$ to $A'$, apply it to $B$, and retrieve the image $B'$ that completes the analogy. It requires the model abstracts an implicit transformation rule from one image pair and generalizes it to another, which effectively tests its capacity for high-order visual reasoning. We instantiate this task by converting VASR analogy triplets into a retrieval setting and curated 147 challenging queries.

## 4 EXPERIMENTS

### 4.1 SETTINGS

We evaluated 11 popular embedding models using our $MR^2$-Bench, categorizing them into two main types: text-only embedding models and multimodal embedding models. We employed nDCG@10 as the primary metric, with additional metric results provided in Appendix G.

For *text embedding models*, we assessed two categories: traditional models such as BGE-M3 (Chen et al., 2024) and Qwen3-Embedding (Zhang et al., 2025b), and models optimized for reasoning-

---

[4]https://commons.wikimedia.org/wiki/Category%3AProof_without_words
[5]https://en.wikipedia.org/wiki/Raven%27s_Progressive_Matrices

| Methods | Multimodal Knowledge Retrieval | | | | | | Visual Illustration | | | Visual Relation | | | Avg. |
|---|---|---|---|---|---|---|---|---|---|---|---|---|---|
| | Bio. | Cook. | Gar. | Phy. | Chem. | Earth. | Econ. | Math. | Nat. | Spa. | Puzz. | Ana. | |
| *Text Embedding Models* | | | | | | | | | | | | | |
| BGE-M3 | 18.79 | 12.97 | 12.04 | 14.52 | 6.05 | 16.35 | - | - | - | - | - | - | - |
| *+ Captions* | 34.19 | 24.28 | 17.88 | 21.24 | 9.67 | 25.19 | 45.46 | 9.97 | 23.66 | 9.48 | 0.00 | 3.46 | 18.71 |
| Qwen3 | 23.77 | 20.44 | 12.61 | 17.13 | 8.61 | 19.79 | - | - | - | - | - | - | - |
| *+ Captions* | 29.97 | 29.29 | 18.32 | 21.46 | 9.52 | 23.19 | 49.44 | 21.14 | 26.30 | 9.11 | 0.00 | 4.30 | 20.17 |
| Diver-Emb. | 27.32 | 16.94 | 15.17 | 18.05 | 10.06 | 22.57 | - | - | - | - | - | - | - |
| *+ Captions* | 38.46 | 30.87 | 22.84 | 23.62 | 14.46 | 31.40 | 54.67 | 25.91 | 24.88 | 8.52 | 0.00 | 7.47 | 23.59 |
| BGE-Rea. | 29.01 | 15.37 | 16.31 | 21.00 | 10.62 | 26.20 | - | - | - | - | - | - | - |
| *+ Captions* | 42.60 | 34.40 | 24.94 | 25.61 | 14.31 | 34.57 | 54.31 | 17.16 | 29.86 | 5.52 | 0.00 | 5.88 | 25.35 |
| ReasonIR | 29.85 | 19.72 | 16.22 | 21.56 | 9.83 | 23.56 | - | - | - | - | - | - | - |
| *+ Captions* | 44.75 | 41.91 | 18.79 | 27.33 | 17.45 | 41.22 | **64.04** | **34.49** | 30.70 | 11.65 | 0.00 | 10.89 | 25.72 |
| *Multimodal Embedding Models* | | | | | | | | | | | | | |
| CLIP | 32.85 | 30.57 | 14.06 | 14.86 | 3.50 | 33.23 | 12.97 | 5.64 | 49.34 | 20.89 | 0.19 | 5.09 | 18.59 |
| BGE-VL | 29.41 | 18.36 | 10.50 | 19.51 | 7.12 | 19.73 | 50.80 | 14.31 | 47.97 | 6.46 | 0.00 | 0.75 | 19.53 |
| GME | 34.34 | 39.50 | 19.04 | 19.29 | 7.73 | 28.59 | 36.95 | 7.19 | 39.35 | 15.70 | 0.22 | **11.11** | 21.59 |
| VLM2Vec | 39.37 | 39.38 | 19.87 | 20.28 | 9.03 | 35.71 | 51.44 | 14.16 | 35.06 | 13.94 | 0.62 | 5.85 | 23.72 |
| MM-Emb. | **49.68** | **52.19** | 23.67 | **30.36** | 17.44 | **47.51** | 42.99 | 21.58 | 48.41 | **22.79** | 0.21 | 5.93 | 30.23 |
| Seed-1.6 | 40.64 | 38.12 | **31.77** | 27.91 | **17.80** | 37.17 | 56.13 | 26.10 | **65.16** | 17.29 | **0.93** | 9.21 | **30.68** |

Table 3: **The overall performance of embedding models on MR$^2$-Bench.** We report nDCG@10 for all sub-tasks. Avg. denotes the average score across 12 datasets. The best score on each dataset is shown in bold and the second best is underlined.

intensive retrieval, including ReasonIR (Shao et al., 2025), BGE-Reasoner-Embed[6], and Diver-Embed (Long et al., 2025). We adopted two evaluation approaches for text embedding models: (1) Using only text information from queries and documents, which is limited for tasks where queries or candidates are purely image-based; (2) Replacing images with textual descriptions (captions). For *multimodal embedding models*, we evaluated CLIP (Radford et al., 2021), VISTA (Zhou et al., 2024), BGE-VL (Zhou et al., 2025), MM-Embed (Lin et al., 2024), GME (Zhang et al., 2025a), VLM2VecV2 (Meng et al., 2025), and Seed1.6-Embedding (Seed, 2025). Detailed information on the models and evaluation procedures can be found in Appendix E.

## 4.2 MAIN RESULTS

We summarize the overall evaluation results for all investigated retrieval baselines in MR$^2$-Bench in Table 3. For each sub-task, we report nDCG@10, along with the macro-average (Avg.) across all tasks. All experiments were conducted within each individual sub-task using separate retrieval corpora. Comprehensive evaluation metrics, including Recall@K and MRR@K, can be found in Appendix G. From these results, we draw some primary conclusions:

**1) Current state-of-the-art models underperform on MR$^2$-Bench.** The leading Seed-1.6 Embedding model (Seed, 2025) achieves only 30.68 nDCG@10 on our benchmark. In contrast, it reports 77.78 overall Recall@1 on the popular MMEB leaderboard (Jiang et al., 2025), but its performance drops significantly to 9.91 Recall@1 on MR$^2$-Bench. Additionally, the SOTA reasoning-intensive text retriever, Diver-Retriever (Long et al., 2025), achieves 33.90 nDCG@10 on BRIGHT (Hongjin et al., 2025), yet only reaches 23.59 nDCG@10 on MR$^2$-Bench when evaluated with auxiliary captions. These results highlight the increased challenges posed by our MR$^2$-Bench.

**2) Text retrievers augmented with image captions provide a strong and practical baseline on MR$^2$-Bench.** Since text retrievers cannot directly process images, we replace each image in queries and candidate documents with detailed natural-language descriptions. This augmentation leads to notable improvements. For instance, ReasonIR+*Captions* surpasses popular open-source multimodal retrievers like VLM2Vec-V2 (Meng et al., 2025). On the Stack Exchange subset, adding captions consistently boosts performance across most tasks. These findings confirm that MR$^2$-Bench

---

[6]https://huggingface.co/BAAI/bge-reasoner-embed-qwen3-8b-0923

is fundamentally multimodal, with retrieval performance significantly enhanced by the visual information provided through captions.

**3) Reasoning-oriented text retrievers significantly outperform traditional matching-based retrievers.** Models optimized for reasoning-intensive retrieval, such as ReasonIR and Diver-Retriever, consistently achieve higher nDCG@10 scores on $MR^2$-Bench compared to matching-centric retrievers like BGE-M3 and Qwen3-Embedding. This advantage is evident across various meta-tasks and persists whether visual content is absent or represented as detailed captions. Collectively, these findings suggest that reasoning-oriented capabilities learned in text retrieval effectively transfer to multimodal retrieval tasks requiring complex reasoning.

**4) Multimodal retrievers show potential on $MR^2$-Bench.** Although not specifically designed for reasoning-intensive tasks, multimodal embedding models like MM-Embed and Seed1.6-Embedding lead performance on $MR^2$-Bench. These models notably outperform caption-augmented text retrievers, including those optimized for reasoning. This gap suggests a promising direction for future research in developing reasoning-intensive multimodal retrievers.

**5) Existing methods struggle with capturing complex visual relationships and abstract concepts.** Current models face challenges in effectively perceiving multi-image relationships (Analogy), spatial configurations (Spatial), and abstract graphics (Mathematics, Visual Puzzle). We hypothesize that these difficulties stem from the inherently visual-centric nature of these tasks, which existing embedding models struggle to comprehend fully. Nonetheless, these images are crucial for real-world applications, as their information is difficult to convey through language alone. This indicates substantial potential for future research to enhance multimodal embedding models.

### 4.3 MORE ANALYSIS

#### 4.3.1 THE EFFECTIVENESS OF QUERY REWRITING

**6) Query rewriting enhances both text and multimodal baselines on $MR^2$-Bench.** This generation-augmented retrieval technique clarifies complex user intent and highlights latent constraints, thus facilitating reasoning-intensive retrieval. Although extensively studied in text-only contexts (Gao et al., 2023; Li et al., 2025a), its application to multimodal retrieval remains under-explored. We evaluated a simple, model-agnostic query rewriting pipeline on $MR^2$-Bench. For each query, GPT-5 (OpenAI, 2025) generates step-by-step reasoning, which is then utilized by each retriever (details in Appendix H). As shown in Table 4, both text and multimodal retrievers show notable average improvements. These results indicate that query rewriting is a practical method for enhancing multimodal reasoning-intensive retrieval tasks, consistently improving performance without the need for fine-tuning existing retrievers.

| Methods | Stack Exchange | | | | | | Visual Illustration | | | Visual Relation | | | Avg. |
|---|---|---|---|---|---|---|---|---|---|---|---|---|---|
| | Bio. | Cook. | Gar. | Phy. | Chem. | Earth. | Econ. | Math. | Nat. | Spa. | Puzz. | Ana. | |
| BGE-M3 | 34.19 | 24.28 | 17.88 | 21.24 | 9.67 | 25.19 | 45.46 | 9.97 | 23.66 | 9.48 | 0.00 | 3.46 | 18.71 |
| *+ Rewrite* | 40.41 | 32.94 | 25.66 | 23.12 | 11.98 | 33.63 | 50.88 | 20.09 | 23.38 | 7.13 | 0.00 | 7.91 | **23.09** |
| Seed-1.6 | 40.64 | 38.12 | 31.77 | 27.91 | 17.80 | 37.17 | 56.13 | 26.10 | 65.16 | 17.29 | 0.93 | 9.21 | 30.68 |
| *+ Rewrite* | 41.13 | 41.47 | 37.68 | 29.47 | 20.70 | 42.02 | 50.08 | 30.37 | 65.84 | 31.87 | 1.24 | 14.62 | **33.87** |

Table 4: Performance comparison of BGE-M3 and Seed-1.6 Embedding on $MR^2$-Bench before and after query rewriting, showing significant improvements across most tasks.

#### 4.3.2 THE EFFECTIVENESS OF ADVANCED RERANKING

A common approach to improve retrieval performance is to employ rerankers that jointly process both the query and its retrieved candidates. Existing studies have shown that incorporating an intermediate reasoning step before final scoring can lead to more accurate rankings (Weller et al., 2025; Zhuang et al., 2025; Liu et al., 2025). We also investigate this by incorporating a reranking stage after the initial retrieval on $MR^2$-Bench. Specifically, we test a wide range of rerankers to rerank the top-$k = 20$ candidates retrieved by three base retrievers: Qwen3-Embedding, GME, and Seed-1.6-Embedding. Their retrieved candidates are reanked by: 1) *textual rerankers*: RankLLaMA-7B and RankLLaMA-14B (Ma et al., 2024); 2) *reasoning-enhanced textual rerankers*: Rank1-7B (Weller et al., 2025), RankR1-14B (Zhuang et al., 2025), ReasonRank-32B (Liu et al., 2025), and BGE-

Reasoner-Reranker-32B [7]; 3) *multimodal rerankers*: MonoQwen2-VL-v0.1 (Chaffin & Lac, 2024) and Jina-Reranker-m0 (JinaAI, 2025); and 4) *reasoning-enhanced multimodal rerankers*: Gemma3-27B (Team, 2025), Qwen2.5-VL-72B (Bai et al., 2025), GLM-4.5V (Team et al., 2025), Gemini-2.5-Pro (Comanici et al., 2025), and GPT-5 (OpenAI, 2025). Since there are no off-the-shelf multimodal rerankers that natively support reasoning, we prompt these MLLMs to first perform reasoning and then output a relevance score. Full implementation details are available in Appendix I.1. Average performance based on Seed-1.6-Embedding is shown in Figure 2, and detailed results for all three base retrievers are provided Appendix I.2 and Appendix I.3.

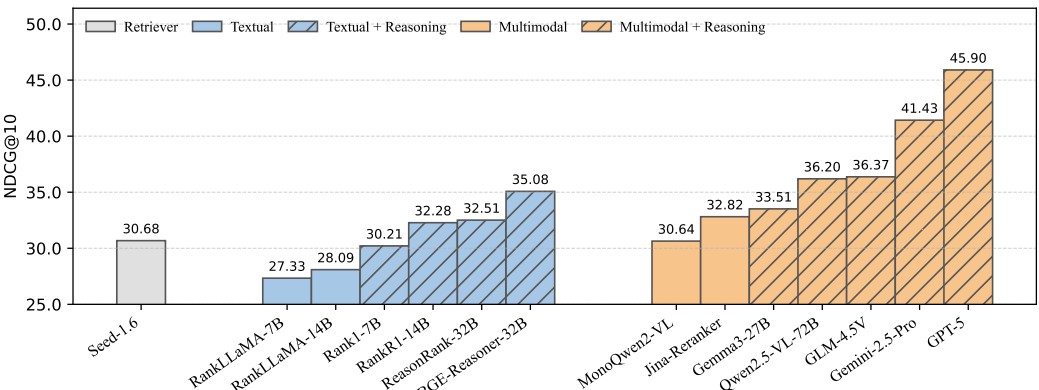

Figure 2: Reranking performance on $MR^2$-Bench with Seed-1.6-Embedding as the base retriever.

From the results presented in Figure 2, we have following findings:

**7) Rerankers deliver substantial gains on $MR^2$-Bench.** Most rerankers significantly outperform the strong Seed-1.6-Embedding baseline, demonstrating the benefit of joint modeling of queries and candidates. Notably, GPT-5 achieves an nDCG@10 of 45.90, an absolute gain of 15.22 over the baseline, indicating the substantial headroom for improvement unlocked by reranking.

**8) An explicit reasoning step before scoring proves to be beneficial.** Across text-only rerankers, those incorporating reasoning consistently outperform their non-reasoning, size-matched counterparts (e.g., Rank1-7B vs. RankLLaMA-7B; RankR1-14B vs. RankLLaMA-14B). This is further substantiated by BGE-Reasoner-Reranker-32B: using only textual input, it achieves an nDCG@10 of 35.08, outperforming the strong base retriever by 4.2 points. Moreover, for multimodal rerankers, models prompted to reason and then rank outperform those trained non-reasoning rerankers. These results confirm that explicit reasoning drives the gains on $MR^2$-Bench.

**9) Multimodal information plays a significant role in enhancing performance.** Despite being built on the lightweight Qwen2-VL-2B backbone, Jina-Reranker-m0 surpasses several larger text-only rerankers, demonstrating clear gains from multimodal information. Furthermore, multimodal models prompted to first reason and then rank (e.g., Qwen2.5-VL-72B, GLM-4.5V, and GPT-5) surpass BGE-Reasoner-Reranker-32B, the best-performing textual reranker specifically trained with reasoning capabilities. GPT-5 achieves the highest overall score, underscoring the importance of utilizing multimodal information with reasoning in tackling the complex retrieval demands posed by $MR^2$-Bench.

## 5 CONCLUSION

In this paper, we introduce $MR^2$-Bench, a novel benchmark for the assessment of multimodal reasoning-intensive retrieval. The comprehensive investigation of existing methods reveals that current retrievers perform poorly on $MR^2$-Bench, with the best models achieving only 30.68 nDCG@10. Our experimental results underscore the importance of multimodal information and reasoning capabilities for effectively addressing $MR^2$-Bench, highlighting significant potential for improvement in this research area. Additionally, we demonstrate that techniques such as query rewriting and reranking can enhance performance on $MR^2$-Bench. We anticipate that this benchmark will facilitate future research in multimodal retrieval, contributing to more realistic and challenging AI applications.

---

[7]https://github.com/FlagOpen/FlagEmbedding/tree/master/research/BGE_Reasoner

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

## APPENDIX

## A USE OF LLMs

In preparing this manuscript, large language models (LLMs) were utilized solely for English grammar checking and polishing. All substantive content and analyses were developed independently by the authors. For dataset construction, GPT-5 (OpenAI, 2025) was employed only for preliminary filtering of candidate data and generating some challenging negative examples, with all final selections and included negative examples thoroughly reviewed and validated by human experts. The relevant procedures are detailed in the appropriate sections of the paper.

## B   ETHICS STATEMENT AND DATA COMPLIANCE

To ensure transparency, legal compliance, and proper re-distribution, we provide a consolidated overview of the data sources, licensing terms, and usage boundaries for all components of MR$^2$-Bench. We confirm that all data collection and redistribution activities strictly adhere to the licenses of the original sources.

### B.1   DATA LICENSING AND USAGE

MR$^2$-Bench integrates data from open platforms and established research datasets. The licensing details for each component are as follows:

**Multimodal Knowledge Retrieval.** The data for this task is derived from *Stack Exchange*, which is licensed under the Creative Commons Attribution-ShareAlike 4.0 International (CC BY-SA 4.0) license. We strictly follow the attribution requirements by preserving metadata links to the original posts. Furthermore, to address privacy concerns, we ensured that user-specific Personally Identifiable Information (PII), such as usernames, avatars, and profile citations, is excluded from the dataset. The data is used solely for academic research purposes.

**Visual Illustration Search.** The economic charts are sourced from *World Bank Open Data*, which is distributed under the CC BY 4.0 license, allowing for redistribution with appropriate attribution. The mathematical visual proofs are curated from *Wikimedia Commons* (Public Domain) and *Proofs Without Words* (used under educational fair use principles), ensuring no copyright infringement for research dissemination.

**Visual Relation and Nature.** We incorporate several public research datasets, all of which allow for academic use and re-distribution. The *INQUIRE* dataset (Nature sub-task) utilizes images from iNaturalist under CC0, CC BY, or CC BY-NC licenses. The *RAVEN* dataset (Visual Puzzle) is licensed under GPL-3.0. The *VASR* dataset (Analogy) operates under the MIT License, and the *CSS* dataset (Spatial) is released under Apache 2.0. Our usage of these datasets is strictly confined to non-commercial research.

### B.2   LICENSE COMPATIBILITY AND DISTRIBUTION

Due to the diverse licensing terms of the constituent sources, MR$^2$-Bench is distributed as a composite dataset.

- **Original Content:** All raw data samples (images and text passages) retain their original licenses as detailed in the section above. Users must strictly adhere to the specific terms of each source.
- **New Contributions:** The benchmark structure, curated queries, and expert annotations created by the authors are released under the CC BY-SA 4.0 license. This ensures compatibility with the ShareAlike requirements of the Stack Exchange data while permitting academic reuse and redistribution of the benchmark's intellectual contributions.

## C   DETAILED OVERVIEW OF MR$^2$-BENCH

We provide detailed modalities of queries and documents, along with the instructions for each sub-task in Table 5. Details on the data sources for each sub-task are provided in Table 6.

## D   MORE DETAILS OF DATA CONSTRUCTION FOR MULTIMODAL KNOWLEDGE RETRIEVAL TASKS

We collected real posts from the Stack Exchange platform to construct our multimodal knowledge retrieval sub-tasks. Queries are derived from actual user questions, while positive documents are sourced from external links in highly voted answers. We utilize BRIGHT's definition to identify a query's positive document: *A document is relevant only if cited in a highly voted answer and confirmed by annotators and domain experts as aiding in reasoning through the query with critical*

| Meta-Task | Sub-Task | Modality $(q \to c)$ | #Queries | #Corpus | Instruction |
|---|---|---|---|---|---|
| MULTIMODAL KNOWLEDGE RETRIEVAL | Biology | $q_{i+t} \to c_{i/t/i+t}$ | 79 | 4,455 | *Find paragraph(s) that could support answering this question.* |
| | Cooking | $q_{i+t} \to c_{i/t/i+t}$ | 76 | 2,786 | *Find paragraph(s) that could support answering this question.* |
| | Gardening | $q_{i+t} \to c_{i/t/i+t}$ | 129 | 5,636 | *Find paragraph(s) that could support answering this question.* |
| | Physics | $q_{i+t} \to c_{i/t/i+t}$ | 76 | 6,656 | *Find paragraph(s) that could support answering this question.* |
| | Chemistry | $q_{i+t} \to c_{i/t/i+t}$ | 124 | 4,317 | *Find paragraph(s) that could support answering this question.* |
| | EarthScience | $q_{i+t} \to c_{i/t/i+t}$ | 99 | 3,014 | *Find paragraph(s) that could support answering this question.* |
| VISUAL ILLUSTRATION SEARCH | Economics | $q_t \to c_i$ | 84 | 7,572 | *Find the chart that best supports answering this question.* |
| | Mathematics | $q_t \to c_i$ | 86 | 944 | *Find the visual proof that best demonstrates this formula.* |
| | Nature | $q_t \to c_i$ | 100 | 2,017 | *Given a natural-world expert query, find the most relevant image.* |
| VISUAL RELATION | Spatial | $q_{i+t} \to c_i$ | 149 | 1,000 | *Given a reference image and a text modification, retrieve the image that best matches the modified reference.* |
| | Visual Puzzle | $q_i \to c_i$ | 160 | 5,375 | *From a 3×3 grid with one missing cell, retrieve the best candidate image to complete the bottom-right cell based on patterns and relations.* |
| | Analogy | $q_i \to c_i$ | 147 | 3,970 | *Given three images, complete the analogy by retrieving the candidate that applies to the third image the relation from the first to the second.* |

Table 5: **The overview of MR$^2$-Bench.** MR$^2$-Bench consists of three meta-tasks and twelve sub-tasks, totaling 1,309 queries. Subscripts indicate the modalities of the query $q$ and candidate $c$: $i$ denotes image, $t$ denotes text, and $i+t$ denotes interleaved image-text.

| META-TASK | Sub-Task | Newly Collected | Source / Adapted From |
|---|---|---|---|
| MULTIMODAL KNOWLEDGE RETRIEVAL | Biology Cooking Gardening Physics Chemistry EarthScience | Yes | Collected from Stack Exchange[2] and external web links. (See Appendix D for details). |
| VISUAL ILLUSTRATION SEARCH | Economics | Yes | Manually collected from World Bank Reports[3] |
| | Mathematics | Yes | Curated from *Proofs Without Words* (Nelsen, 2015) and Wikimedia Commons[4] |
| | Nature | No | Adapted from INQUIRE-Rerank (Vendrow et al., 2024). |
| VISUAL RELATION REASONING | Spatial | No | Adapted from CSS dataset (Vo et al., 2019). |
| | Visual Puzzle | No | Reorganized from RAVEN dataset (Zhang et al., 2019). |
| | Analogy | No | Adapted from VASR dataset (Bitton et al., 2023). |

Table 6: **Data sources for MR$^2$-Bench.** We specify whether each sub-task was newly collected for this benchmark or adapted from existing datasets.

*concepts or theories* (Hongjin et al., 2025). Given the multimodal nature of the task in MR²-Bench, our annotation process diverges from BRIGHT's construction methodology. The specific steps of our process are summarized as follows:

**Initial Posts Collection and Filtering.** We initiated the process by gathering a substantial set of posts from Stack Exchange. To ensure data quality and relevance, we retained posts meeting specific criteria: (1) the question must contain image(s) essential for understanding the query; (2) the post must have received at least five community votes, indicating reliability; and (3) the answer must include at least one external link to facilitate further content acquisition.

**Web Page Acquisition and Paragraph Annotation.** For each qualifying post, annotators are required to visit the external links provided in the answers and copy the interleaved text-image content in the order it appears, excluding Wikipedia.[8] They then segment this content into paragraphs, preserving images to maintain multimodal information. This process generates a collection of candidate paragraphs for each query, including both text-only and image-containing segments. Initial identification of positive paragraphs is performed using GPT-5 (OpenAI, 2025), followed by expert validation to ensure accuracy and relevance. Only queries with at least one confirmed positive paragraph are included in the final dataset.

**Incorporation of Challenging Negative Examples.** To rigorously assess the reasoning capabilities of evaluation methods, we introduced challenging negative samples for each retained query using two strategies: (1) retrieving topic-related documents from an internal corpus using the query's keywords, with GPT-5 initially verifying they are not false negatives; and (2) using GPT-5 to generate documents that, while topically related, provide unhelpful information. All negative samples were subsequently reviewed by human experts to ensure the integrity of the benchmark.

## E    MORE DETAILS OF BASELINES

In our evaluation, we classify the retriever baseline into two main categories: text embedding models and multimodal embedding models. We assess the Seed1.6-Embedding model (Seed, 2025) via its official API, whereas all other models are evaluated using their publicly available code and open-source checkpoints. Below, we provide a comprehensive overview of the implementation details for all baselines used in the evaluation process.

### E.1    TEXT EMBEDDING MODELS

The evaluated text retrievers include: BGE-M3 (Chen et al., 2024), Qwen3-Embedding (Zhang et al., 2025b), ReasonIR (Shao et al., 2025), BGR-Reasoner-Embed[9], and Diver-Embed (Long et al., 2025). Notably, the last three models have been fine-tuned specifically for reasoning-intensive retrieval tasks, as detailed in their technical reports or repository descriptions.

We consider two input configurations for all text-only retrievers. The first configuration ignores images, utilizing only the textual content from queries and documents; this setup is not applicable to some sub-tasks where either the query or candidates are purely visual. The second configuration employs a caption-augmented approach, where every image in both queries and documents is replaced with a textual description. Specifically, we use the Qwen2.5-VL-7B model (Bai et al., 2025) to generate captions for the images with the prompt: *Write a detailed English caption for this image, covering the main objects, their attributes, relationships, actions, layout, and background elements.* Each image in the original input is then substituted with a caption prefixed by its identifier, formatted as `[IMAGE_id]:   image_caption`.

### E.2    MULTIMODAL EMBEDDING MODELS

The evaluated multimodal retrievers include CLIP (Radford et al., 2021), BGE-VL (Zhou et al., 2025), GME (Zhang et al., 2025a), VLM2Vec-V2 (Meng et al., 2025), MM-Embed (Lin et al., 2024), and Seed1.6-Embedding (Seed, 2025). All these models can process individual images and

---

[8]Wikipedia content was automatically extracted using Playwright to minimize manual effort.
[9]https://huggingface.co/BAAI/bge-reasoner-embed-qwen3-8b-0923

texts directly. However, for interleaved image-text data with multiple images, different models require specific handling approaches:

For the CLIP model, we employ a score fusion strategy, following previous work (Wei et al., 2024). This involves separately embedding the image and text data and then combining these embeddings through element-wise addition to achieve the final image-text representation.

For models that can only input a single image in image-text data, specifically BGE-VL (Zhou et al., 2025) and MM-Embed (Lin et al., 2024), we create a composite image by tiling multiple images together, which is then processed jointly with the text.

For other models capable of handling interleaved image-text data with multiple images, we preserve the sequence of images and text, allowing their processors to generate interleaved image-text tokens, which are then used to derive the final embeddings.

## F    SENSITIVITY ANALYSIS OF CAPTIONING MODELS

To investigate whether the performance of text-based retrievers on $MR^2$-Bench is sensitive to the choice of the captioning model, we conducted a comprehensive sensitivity analysis using the state-of-the-art multimodal language model GLM-4.1V-9B-Thinking [10], which is known for its chain-of-thought reasoning capabilities. The goal was to verify that the observed results are not biased by the specific captioning model (Qwen2.5-VL-7B) used in the main experiments.

In this analysis, we replaced the original captions with those generated by GLM-4.1V-9B-Thinking using the same prompt structure and re-evaluated four representative text embedding models: BGE-M3, Qwen3-Embedding, Diver-Retriever, and ReasonIR. A detailed comparison across all 12 sub-tasks is presented in Table 7.

The results show that the relative performance ranking of the retrievers remains consistent regardless of the captioning model used. Specifically, the reasoning-enhanced retrievers (ReasonIR and Diver-Retriever) consistently outperform the standard retrievers (Qwen3-Embedding and BGE-M3), confirming that our main findings are robust and not artifacts of caption style alignment.

Different captioning models do exhibit varying strengths depending on the domain. For instance, captions generated by GLM-4.1V-9B-Thinking resulted in performance gains in the *Economics* domain (e.g., Diver-Retriever improved from 54.67 to 56.35), likely due to more detailed chart descriptions provided by this model. In contrast, for the *Mathematics* domain, Qwen2.5-VL captions proved slightly more effective for certain retrievers. However, despite these domain-specific variations, the Macro-Average scores across all 12 sub-tasks remain comparable (e.g., ReasonIR: 25.72 vs. 26.26), demonstrating the stability and consistency of the benchmark metrics.

## G    DETAILED EVALUATION METRICS OF $MR^2$-BENCH

In this section, we provide more detailed evaluation results of the embedding models on $MR^2$-Bench. Table 8, Table 9, Table 10, Table 11, Table 12, Table 13, and Table 14 present the performance of the embedding models in terms of Recall@1, Recall@5, Recall@10, nDCG@5, nDCG@20, MRR@5, MRR@10.

## H    MORE DETAILS OF IMPLEMENTATION FOR QUERY REWRITING

Given the strong reasoning capabilities of Multimodal Large Language Models (MLLMs), we take advantage of their ability to produce explicit step-by-step chain-of-thought reasoning in order to improve the effectiveness of query rewriting and thereby enhance retrieval performance. Instead of relying on a single direct reformulation, we design a prompting strategy that guides the MLLM through a structured reasoning process. Concretely, the model is first asked to (i) identify the most salient subquestions that are implicitly contained in the given instruction and query, ensuring that complex or multifaceted information needs are decomposed into clear components. Next, the model is prompted to (ii) reason step-by-step about what types of evidence, textual patterns, and document

---

[10]https://huggingface.co/zai-org/GLM-4.1V-9B-Thinking

| Methods | Multimodal Knowledge Retrieval | | | | | | Visual Illustration | | | Visual Relation | | | Avg. |
|---|---|---|---|---|---|---|---|---|---|---|---|---|---|
| | Bio. | Cook. | Gar. | Phy. | Chem. | Earth. | Econ. | Math. | Nat. | Spa. | Puzz. | Ana. | |
| BGE-M3 | 18.79 | 12.97 | 12.04 | 14.52 | 6.05 | 16.35 | - | - | - | - | - | - | - |
| + *Captions (Qwen)* | 34.19 | 24.28 | 17.88 | 21.24 | 9.67 | 25.19 | 45.46 | 9.97 | 23.66 | 9.48 | 0.00 | 3.46 | 18.71 |
| + *Captions (GLM)* | 35.56 | 26.27 | 16.11 | 22.64 | 9.91 | 24.39 | 39.58 | 9.15 | 24.22 | 10.36 | 0.00 | 4.91 | 18.59 |
| Qwen3 | 23.77 | 20.44 | 12.61 | 17.13 | 8.61 | 19.79 | - | - | - | - | - | - | - |
| + *Captions (Qwen)* | 29.97 | 29.29 | 18.32 | 21.46 | 9.52 | 23.19 | 49.44 | 21.14 | 26.30 | 9.11 | 0.00 | 4.30 | 20.17 |
| + *Captions (GLM)* | 30.26 | 29.50 | 15.54 | 20.54 | 9.30 | 22.89 | 43.89 | 16.42 | 28.31 | 6.69 | 0.00 | 8.45 | 19.32 |
| Diver-Emb. | 27.32 | 16.94 | 15.17 | 18.05 | 10.06 | 22.57 | - | - | - | - | - | - | - |
| + *Captions (Qwen)* | 38.46 | 30.87 | 22.84 | 23.62 | 14.46 | 31.40 | 54.67 | 25.91 | 24.88 | 8.52 | 0.00 | 7.47 | 23.59 |
| + *Captions (GLM)* | 39.70 | 32.17 | 21.59 | 24.13 | 14.22 | 32.36 | 56.35 | 28.32 | 28.44 | 7.48 | 0.00 | 7.81 | 24.38 |
| ReasonIR | 29.85 | 19.72 | 16.22 | 21.56 | 9.83 | 23.56 | - | - | - | - | - | - | - |
| + *Captions (Qwen)* | 44.75 | 41.91 | 18.79 | 27.33 | 17.45 | 41.22 | 64.04 | 34.49 | 30.70 | 11.65 | 0.00 | 10.89 | 25.72 |
| + *Captions (GLM)* | 46.06 | 42.02 | 20.24 | 26.79 | 17.47 | 38.52 | 57.49 | 13.81 | 33.49 | 10.63 | 0.00 | 8.63 | 26.26 |

Table 7: **Sensitivity Analysis of Captioning Models.** Comparing performance (nDCG@10) of text retrievers augmented with captions generated by Qwen2.5-VL-7B (Gray rows) versus GLM-4v-9B-Thinking (Blue rows). The results demonstrate that while absolute scores fluctuate across sub-tasks due to different captioning styles, the relative ranking of retrieval models remains highly consistent.

| Methods | Multimodal Knowledge Retrieval | | | | | | Visual Illustration | | | Visual Relation | | | Avg. |
|---|---|---|---|---|---|---|---|---|---|---|---|---|---|
| | Bio. | Cook. | Gar. | Phy. | Chem. | Earth. | Econ. | Math. | Nat. | Spa. | Puzz. | Ana. | |
| | *Text Embedding Models* | | | | | | | | | | | | |
| BGE-M3 | 3.61 | 2.25 | 3.26 | 2.70 | 1.04 | 3.77 | - | - | - | - | - | - | - |
| + *Captions* | 10.22 | 3.23 | 6.44 | 5.00 | 1.43 | 7.26 | 32.14 | 3.49 | 6.67 | 4.00 | 0.00 | 1.36 | 6.77 |
| Qwen3 | 5.46 | 2.67 | 3.11 | 1.86 | 1.13 | 4.67 | - | - | - | - | - | - | - |
| + *Captions* | 7.21 | 6.87 | 5.15 | 4.84 | 1.20 | 5.93 | 32.14 | 6.98 | 4.92 | 4.03 | 0.00 | 0.68 | 6.66 |
| Diver-Emb. | 5.73 | 2.55 | 4.74 | 1.60 | 0.38 | 3.71 | - | - | - | - | - | - | - |
| + *Captions* | 12.37 | 7.06 | 9.87 | 4.60 | 2.39 | 6.60 | 36.90 | 8.14 | 3.00 | 3.33 | 0.00 | 0.68 | 7.91 |
| BGE-Rea. | 3.69 | 3.13 | 4.10 | 2.59 | 1.36 | 4.64 | - | - | - | - | - | - | - |
| + *Captions* | 16.03 | 9.84 | 9.74 | 6.19 | 1.24 | 10.28 | 41.67 | 12.21 | 1.67 | 2.00 | 0.00 | 0.68 | 9.29 |
| ReasonIR | 7.68 | 3.13 | 3.75 | 4.35 | 0.91 | 4.21 | - | - | - | - | - | - | - |
| + *Captions* | 16.87 | 13.81 | 7.13 | 5.32 | 3.50 | 11.59 | 39.29 | 7.56 | 6.58 | 2.00 | 0.00 | 0.68 | 9.53 |
| | *Multimodal Embedding Models* | | | | | | | | | | | | |
| CLIP | 12.49 | 8.28 | 4.37 | 2.58 | 1.42 | 11.72 | 3.57 | 1.16 | 10.92 | 12.67 | 0.00 | 0.00 | 5.77 |
| BGE-VL | 8.96 | 2.30 | 2.93 | 4.35 | 0.32 | 4.81 | 34.52 | 6.98 | 10.83 | 2.01 | 0.00 | 1.36 | 6.62 |
| GME | 10.07 | 14.48 | 7.84 | 3.97 | 1.43 | 8.39 | 21.43 | 2.33 | 8.08 | 8.00 | 0.00 | 3.40 | 7.45 |
| VLM2Vec | 13.58 | 13.73 | 5.41 | 3.73 | 1.44 | 14.54 | 38.10 | 3.49 | 9.53 | 4.00 | 0.62 | 0.68 | 9.07 |
| MM-Emb. | 17.18 | 20.81 | 7.10 | 7.05 | 4.35 | 17.54 | 34.52 | 9.59 | 11.25 | 11.33 | 0.00 | 0.00 | 11.73 |
| Seed-1.6 | 13.65 | 9.02 | 9.85 | 5.20 | 3.69 | 9.81 | 33.33 | 6.98 | 19.33 | 8.00 | 0.00 | 0.00 | 9.91 |

Table 8: The overall performance of embedding models on $MR^2$-Bench in terms of the Recall@1.

attributes would be necessary for relevant sources to contain, which encourages a more targeted and discriminative retrieval process. Finally, model (iii) produces both an explicit reasoning trace, which captures its internal deliberation, and a set of candidate rewritten queries or answers that can be used to drive retrieval more effectively. We employ GPT-5 (OpenAI, 2025), the SOTA multimodal reasoning model, to perform query rewriting. The prompt is provided in Figure 3.

# I   MORE DETAILS OF RERANKING

## I.1   IMPLEMENTATION DETAILS

For text-only rerankers, following the second input configuration described in Section E, we append image captions as auxiliary context. For multimodal rerankers, MLLMs are prompted in a

| Methods | Multimodal Knowledge Retrieval | | | | | | Visual Illustration | | | Visual Relation | | | Avg. |
|---|---|---|---|---|---|---|---|---|---|---|---|---|---|
| | Bio. | Cook. | Gar. | Phy. | Chem. | Earth. | Econ. | Math. | Nat. | Spa. | Puzz. | Ana. | |
| *Text Embedding Models* | | | | | | | | | | | | | |
| BGE-M3 | 14.09 | 10.43 | 11.66 | 10.12 | 7.27 | 12.78 | - | - | - | - | - | - | - |
| *+ Captions* | 28.85 | 23.21 | 17.32 | 13.60 | 8.76 | 20.66 | 53.57 | 10.85 | 20.25 | 11.33 | 0.00 | 3.40 | 17.65 |
| Qwen3 | 17.36 | 12.48 | 12.17 | 11.93 | 6.38 | 15.12 | - | - | - | - | - | - | - |
| *+ Captions* | 24.76 | 27.10 | 16.30 | 13.73 | 6.61 | 17.95 | 60.71 | 30.33 | 24.42 | 11.41 | 0.00 | 5.44 | 19.90 |
| Diver-Emb. | 24.54 | 12.43 | 15.95 | 13.61 | 8.07 | 19.33 | - | - | - | - | - | - | - |
| *+ Captions* | 30.82 | 27.00 | 20.75 | 16.47 | 11.35 | 29.76 | 65.48 | 37.50 | 21.67 | 10.00 | 0.00 | 10.88 | 23.47 |
| BGE-Rea. | 23.36 | 8.80 | 15.83 | 13.79 | 8.05 | 19.01 | - | - | - | - | - | - | - |
| *+ Captions* | 33.49 | 32.50 | 25.09 | 17.00 | 12.56 | 26.61 | 70.24 | 46.71 | 25.42 | 8.67 | 0.00 | 6.80 | 25.42 |
| ReasonIR | 26.10 | 16.55 | 14.73 | 14.94 | 10.08 | 20.38 | - | - | - | - | - | - | - |
| *+ Captions* | 33.01 | 36.37 | 16.49 | 20.05 | 17.45 | 36.66 | 61.90 | 20.93 | 24.33 | 6.00 | 0.00 | 8.16 | 23.45 |
| *Multimodal Embedding Models* | | | | | | | | | | | | | |
| CLIP | 27.54 | 28.63 | 9.72 | 7.60 | 4.01 | 29.62 | 16.67 | 4.65 | 48.17 | 22.67 | 0.00 | 6.80 | 17.17 |
| BGE-VL | 22.17 | 12.55 | 10.96 | 15.18 | 6.30 | 15.22 | 63.10 | 17.64 | 47.33 | 10.07 | 0.00 | 5.44 | 18.83 |
| GME | 27.06 | 33.94 | 15.53 | 13.36 | 6.25 | 24.78 | 45.24 | 6.40 | 37.42 | 20.67 | 0.00 | 12.24 | 20.24 |
| VLM2Vec | 30.64 | 34.61 | 18.89 | 12.44 | 7.36 | 32.18 | 55.95 | 17.25 | 31.75 | 17.33 | 0.63 | 4.08 | 21.93 |
| MM-Emb. | 38.24 | 48.89 | 21.07 | 21.03 | 16.53 | 42.79 | 48.81 | 22.58 | 42.08 | 28.00 | 0.00 | 6.12 | 28.01 |
| Seed-1.6 | 31.93 | 32.51 | 28.95 | 22.17 | 14.52 | 31.65 | 69.05 | 38.76 | 61.25 | 19.33 | 0.63 | 8.16 | 29.91 |

Table 9: The overall performance of embedding models on MR$^2$-Bench in terms of the Recall@5.

| Methods | Multimodal Knowledge Retrieval | | | | | | Visual Illustration | | | Visual Relation | | | Avg. |
|---|---|---|---|---|---|---|---|---|---|---|---|---|---|
| | Bio. | Cook. | Gar. | Phy. | Chem. | Earth. | Econ. | Math. | Nat. | Spa. | Puzz. | Ana. | |
| *Text Embedding Models* | | | | | | | | | | | | | |
| BGE-M3 | 25.92 | 18.48 | 17.29 | 17.00 | 9.11 | 22.31 | - | - | - | - | - | - | - |
| *+ Captions* | 39.67 | 35.42 | 20.81 | 21.23 | 14.02 | 32.55 | 61.90 | 19.57 | 33.83 | 16.67 | 0.00 | 7.48 | 25.26 |
| Qwen3 | 32.83 | 30.81 | 17.08 | 20.64 | 13.94 | 27.05 | - | - | - | - | - | - | - |
| *+ Captions* | 33.91 | 38.52 | 24.39 | 21.03 | 15.41 | 31.27 | 67.86 | 38.08 | 39.67 | 16.11 | 0.00 | 8.84 | 27.92 |
| Diver-Emb. | 35.18 | 25.13 | 20.01 | 22.07 | 16.42 | 33.76 | - | - | - | - | - | - | - |
| *+ Captions* | 43.81 | 42.45 | 26.21 | 25.29 | 22.66 | 43.71 | 70.24 | 43.31 | 40.83 | 16.00 | 0.00 | 15.65 | 32.51 |
| BGE-Rea. | 41.29 | 24.18 | 23.80 | 23.82 | 17.24 | 37.44 | - | - | - | - | - | - | - |
| *+ Captions* | 46.35 | 42.84 | 29.78 | 24.51 | 22.96 | 43.02 | 79.76 | 58.53 | 40.42 | 12.00 | 0.00 | 11.56 | 34.31 |
| ReasonIR | 37.42 | 29.45 | 22.93 | 24.88 | 16.21 | 34.28 | - | - | - | - | - | - | - |
| *+ Captions* | 46.05 | 50.89 | 20.99 | 28.93 | 25.72 | 52.31 | 69.05 | 28.88 | 44.08 | 10.00 | 0.00 | 12.93 | 32.48 |
| *Multimodal Embedding Models* | | | | | | | | | | | | | |
| CLIP | 33.08 | 38.08 | 14.38 | 14.05 | 5.64 | 38.85 | 26.19 | 12.21 | 70.42 | 31.33 | 0.63 | 11.56 | 24.70 |
| BGE-VL | 38.03 | 27.68 | 16.66 | 22.85 | 11.62 | 28.46 | 66.67 | 23.45 | 67.83 | 12.08 | 0.00 | 12.93 | 27.35 |
| GME | 35.13 | 45.64 | 21.93 | 19.66 | 13.14 | 36.10 | 54.76 | 15.50 | 57.17 | 25.33 | 0.63 | 23.13 | 29.01 |
| VLM2Vec | 41.02 | 44.31 | 23.69 | 20.66 | 13.20 | 39.49 | 66.67 | 27.23 | 47.67 | 27.33 | 0.63 | 14.97 | 30.57 |
| MM-Emb. | 50.98 | 55.18 | 26.60 | 28.91 | 22.79 | 54.61 | 51.19 | 35.08 | 68.42 | 35.33 | 0.63 | 14.29 | 37.00 |
| Seed-1.6 | 47.99 | 49.13 | 38.60 | 30.05 | 26.32 | 48.90 | 79.76 | 47.87 | 84.17 | 30.67 | 2.50 | 22.45 | 42.37 |

Table 10: The overall performance of embedding models on MR$^2$-Bench in terms of the Recall@10.

| Methods | Multimodal Knowledge Retrieval | | | | | | Visual Illustration | | | Visual Relation | | | Avg. |
|---|---|---|---|---|---|---|---|---|---|---|---|---|---|
| | Bio. | Cook. | Gar. | Phy. | Chem. | Earth. | Econ. | Math. | Nat. | Spa. | Puzz. | Ana. | |
| *Text Embedding Models* | | | | | | | | | | | | | |
| BGE-M3 | 14.89 | 10.15 | 9.78 | 13.18 | 5.33 | 13.23 | - | - | - | - | - | - | - |
| *+ Captions* | 32.01 | 19.33 | 17.27 | 21.47 | 8.02 | 21.21 | 42.82 | 7.13 | 17.66 | 7.81 | 0.00 | 2.18 | 16.41 |
| Qwen3 | 18.71 | 13.35 | 10.80 | 14.72 | 5.76 | 15.71 | - | - | - | - | - | - | - |
| *+ Captions* | 27.36 | 24.45 | 15.25 | 20.60 | 6.75 | 18.77 | 47.01 | 18.56 | 19.48 | 7.70 | 0.00 | 3.20 | 17.43 |
| Diver-Emb. | 24.49 | 11.88 | 13.45 | 16.24 | 6.70 | 17.47 | - | - | - | - | - | - | - |
| *+ Captions* | 36.03 | 24.59 | 20.72 | 21.95 | 10.66 | 26.57 | 53.13 | 23.94 | 16.49 | 6.60 | 0.00 | 5.97 | 20.56 |
| BGE-Rea. | 23.63 | 8.90 | 12.79 | 18.98 | 6.92 | 19.62 | - | - | - | - | - | - | - |
| *+ Captions* | 40.07 | 30.43 | 23.44 | 25.26 | 10.06 | 28.38 | 57.44 | 30.40 | 18.43 | 5.59 | 0.00 | 4.11 | 22.80 |
| ReasonIR | 26.90 | 14.77 | 13.12 | 19.15 | 7.68 | 17.96 | - | - | - | - | - | - | - |
| *+ Captions* | 42.83 | 36.92 | 17.98 | 26.60 | 14.74 | 36.36 | 52.01 | 14.34 | 21.23 | 4.30 | 0.00 | 4.36 | 22.64 |
| *Multimodal Embedding Models* | | | | | | | | | | | | | |
| CLIP | 33.19 | 27.83 | 10.42 | 15.34 | 4.90 | 30.50 | 9.91 | 3.13 | 39.38 | 18.14 | 0.00 | 3.57 | 16.36 |
| BGE-VL | 26.00 | 12.99 | 9.92 | 18.10 | 5.71 | 14.74 | 49.60 | 12.44 | 39.07 | 5.83 | 0.00 | 3.53 | 16.49 |
| GME | 33.91 | 35.58 | 17.03 | 18.41 | 5.46 | 25.61 | 33.89 | 4.18 | 30.64 | 14.21 | 0.00 | 7.64 | 18.88 |
| VLM2Vec | 38.31 | 36.87 | 18.75 | 19.66 | 7.46 | 34.05 | 47.87 | 10.86 | 28.12 | 10.74 | 0.63 | 2.46 | 21.31 |
| MM-Emb. | 48.80 | 50.58 | 22.22 | 30.84 | 15.50 | 44.52 | 42.15 | 17.22 | 37.07 | 20.34 | 0.00 | 3.29 | 27.71 |
| Seed-1.6 | 36.14 | 32.45 | 28.34 | 27.69 | 13.46 | 31.52 | 52.63 | 22.95 | 55.12 | 13.67 | 0.31 | 4.49 | 26.56 |

Table 11: The overall performance of embedding models on MR$^2$-Bench in terms of the nDCG@5.

| Methods | Multimodal Knowledge Retrieval | | | | | | Visual Illustration | | | Visual Relation | | | Avg. |
|---|---|---|---|---|---|---|---|---|---|---|---|---|---|
| | Bio. | Cook. | Gar. | Phy. | Chem. | Earth. | Econ. | Math. | Nat. | Spa. | Puzz. | Ana. | |
| *Text Embedding Models* | | | | | | | | | | | | | |
| BGE-M3 | 22.66 | 16.33 | 13.67 | 17.72 | 7.89 | 19.84 | - | - | - | - | - | - | - |
| *+ Captions* | 37.22 | 28.43 | 19.70 | 24.36 | 11.39 | 30.06 | 47.59 | 12.65 | 27.18 | 10.95 | 0.00 | 5.14 | 21.22 |
| Qwen3 | 30.34 | 24.25 | 15.39 | 20.39 | 12.16 | 24.16 | - | - | - | - | - | - | - |
| *+ Captions* | 36.76 | 33.24 | 21.41 | 24.91 | 13.21 | 29.03 | 49.75 | 24.00 | 30.48 | 10.96 | 0.00 | 6.70 | 23.37 |
| Diver-Emb. | 32.45 | 23.00 | 17.59 | 24.00 | 13.65 | 28.12 | - | - | - | - | - | - | - |
| *+ Captions* | 43.90 | 36.21 | 25.50 | 28.19 | 18.26 | 37.13 | 58.30 | 29.83 | 29.68 | 10.52 | 0.17 | 8.86 | 27.21 |
| BGE-Rea. | 33.43 | 21.87 | 18.97 | 25.40 | 14.43 | 30.65 | - | - | - | - | - | - | - |
| *+ Captions* | 47.04 | 39.94 | 28.24 | 29.94 | 19.20 | 40.59 | 63.25 | 35.94 | 33.36 | 8.87 | 0.00 | 7.89 | 29.52 |
| ReasonIR | 36.90 | 24.69 | 18.92 | 25.97 | 13.12 | 30.39 | - | - | - | - | - | - | - |
| *+ Captions* | 48.18 | 45.34 | 21.83 | 28.71 | 21.15 | 44.74 | 57.61 | 19.33 | 36.48 | 6.19 | 0.00 | 8.91 | 28.21 |
| *Multimodal Embedding Models* | | | | | | | | | | | | | |
| CLIP | 35.49 | 31.94 | 13.96 | 16.53 | 6.01 | 34.38 | 14.76 | 6.57 | 56.32 | 23.04 | 0.53 | 6.71 | 20.52 |
| BGE-VL | 36.96 | 26.11 | 17.09 | 23.25 | 9.47 | 26.61 | 52.91 | 16.12 | 53.97 | 7.71 | 0.17 | 8.46 | 23.24 |
| GME | 38.17 | 43.48 | 20.72 | 21.56 | 10.82 | 33.43 | 40.88 | 9.08 | 45.85 | 18.74 | 0.22 | 14.88 | 24.82 |
| VLM2Vec | 42.11 | 43.24 | 21.36 | 21.93 | 11.63 | 38.91 | 55.66 | 18.46 | 40.38 | 17.18 | 0.63 | 8.77 | 26.69 |
| MM-Emb. | 51.83 | 54.36 | 26.38 | 32.74 | 20.39 | 51.77 | 45.99 | 22.91 | 55.04 | 23.97 | 0.36 | 8.00 | 32.81 |
| Seed-1.6 | 46.01 | 43.31 | 35.86 | 32.99 | 22.85 | 43.71 | 58.25 | 28.38 | 69.97 | 21.20 | 1.67 | 11.76 | 34.66 |

Table 12: The overall performance of embedding models on MR$^2$-Bench in terms of the nDCG@20.

| Methods | Multimodal Knowledge Retrieval | | | | | | Visual Illustration | | | Visual Relation | | | Avg. |
|---|---|---|---|---|---|---|---|---|---|---|---|---|---|
| | Bio. | Cook. | Gar. | Phy. | Chem. | Earth. | Econ. | Math. | Nat. | Spa. | Puzz. | Ana. | |
| *Text Embedding Models* | | | | | | | | | | | | | |
| BGE-M3 | 21.05 | 14.78 | 13.28 | 23.18 | 6.96 | 19.29 | - | - | - | - | - | - | 16.43 |
| *+ Captions* | 42.95 | 23.84 | 25.36 | 31.49 | 12.11 | 29.36 | 39.31 | 6.38 | 27.10 | 6.66 | 0.00 | 1.80 | 20.53 |
| Qwen3 | 34.51 | 24.17 | 21.24 | 25.35 | 13.24 | 20.13 | - | - | - | - | - | - | 23.11 |
| *+ Captions* | 47.53 | 31.58 | 25.23 | 29.14 | 14.61 | 26.43 | 56.65 | 27.97 | 30.00 | 8.54 | 0.00 | 5.69 | 25.28 |
| Diver-Emb. | 32.03 | 17.08 | 17.69 | 21.86 | 8.95 | 20.93 | - | - | - | - | - | - | 19.76 |
| *+ Captions* | 44.22 | 31.64 | 28.86 | 29.71 | 14.57 | 31.62 | 48.91 | 20.25 | 23.23 | 5.49 | 0.00 | 4.35 | 23.57 |
| BGE-Rea. | 28.40 | 11.86 | 16.43 | 26.91 | 10.82 | 25.29 | - | - | - | - | - | - | 19.95 |
| *+ Captions* | 51.03 | 35.83 | 30.54 | 34.76 | 14.14 | 37.73 | 53.13 | 25.64 | 24.43 | 4.56 | 0.00 | 3.20 | 26.25 |
| ReasonIR | 33.04 | 19.91 | 17.20 | 28.07 | 9.78 | 22.56 | - | - | - | - | - | - | 21.76 |
| *+ Captions* | 55.15 | 46.27 | 25.76 | 35.44 | 18.00 | 46.89 | 48.65 | 12.93 | 31.17 | 3.72 | 0.00 | 3.13 | 27.26 |
| *Multimodal Embedding Models* | | | | | | | | | | | | | |
| CLIP | 44.64 | 35.07 | 15.09 | 25.19 | 8.65 | 39.72 | 7.72 | 2.62 | 46.75 | 16.61 | 0.00 | 2.51 | 20.38 |
| BGE-VL | 36.31 | 18.44 | 14.12 | 25.29 | 6.96 | 19.02 | 45.14 | 10.68 | 46.47 | 4.46 | 0.00 | 2.89 | 19.15 |
| GME | 46.18 | 45.07 | 24.60 | 26.51 | 6.56 | 34.53 | 30.14 | 3.59 | 38.55 | 12.13 | 0.00 | 6.16 | 22.83 |
| VLM2Vec | 49.49 | 48.77 | 25.89 | 30.13 | 9.88 | 42.32 | 45.16 | 9.07 | 36.57 | 8.59 | 0.62 | 1.93 | 25.70 |
| MM-Emb. | 60.00 | 59.52 | 32.78 | 40.07 | 20.50 | 53.43 | 39.92 | 17.09 | 47.77 | 17.79 | 0.00 | 2.35 | 32.60 |
| Seed-1.6 | 45.38 | 41.54 | 38.45 | 34.01 | 18.32 | 39.34 | 47.16 | 18.24 | 63.07 | 11.82 | 0.21 | 3.25 | 30.07 |

Table 13: The overall performance of embedding models on MR$^2$-Bench in terms of the MRR@5.

| Methods | Multimodal Knowledge Retrieval | | | | | | Visual Illustration | | | Visual Relation | | | Avg. |
|---|---|---|---|---|---|---|---|---|---|---|---|---|---|
| | Bio. | Cook. | Gar. | Phy. | Chem. | Earth. | Econ. | Math. | Nat. | Spa. | Puzz. | Ana. | |
| *Text Embedding Models* | | | | | | | | | | | | | |
| BGE-M3 | 23.01 | 16.21 | 14.34 | 25.04 | 7.90 | 21.36 | - | - | - | - | - | - | 17.98 |
| *+ Captions* | 44.71 | 27.02 | 25.84 | 32.19 | 13.10 | 31.09 | 40.38 | 7.60 | 28.41 | 7.32 | 0.00 | 2.32 | 21.67 |
| Qwen3 | 36.73 | 25.96 | 22.44 | 26.47 | 15.38 | 22.63 | - | - | - | - | - | - | 24.93 |
| *+ Captions* | 48.98 | 32.87 | 26.38 | 30.26 | 16.47 | 28.40 | 58.34 | 29.39 | 31.81 | 9.33 | 0.00 | 7.09 | 26.61 |
| Diver-Emb. | 33.64 | 19.73 | 18.39 | 23.02 | 10.64 | 23.16 | - | - | - | - | - | - | 21.43 |
| *+ Captions* | 45.47 | 34.17 | 30.00 | 31.56 | 16.79 | 33.39 | 49.55 | 21.12 | 26.03 | 6.28 | 0.00 | 4.95 | 24.94 |
| BGE-Rea. | 31.03 | 15.53 | 18.25 | 28.22 | 13.09 | 27.92 | - | - | - | - | - | - | 22.34 |
| *+ Captions* | 52.27 | 37.38 | 31.21 | 35.57 | 15.88 | 39.88 | 54.32 | 27.52 | 26.42 | 4.96 | 0.00 | 3.88 | 27.44 |
| ReasonIR | 35.35 | 21.68 | 18.46 | 29.46 | 10.77 | 24.87 | - | - | - | - | - | - | 23.44 |
| *+ Captions* | 56.61 | 48.19 | 26.44 | 36.65 | 19.59 | 48.36 | 49.60 | 14.27 | 34.01 | 4.18 | 0.00 | 3.75 | 28.47 |
| *Multimodal Embedding Models* | | | | | | | | | | | | | |
| CLIP | 45.80 | 36.53 | 16.04 | 26.33 | 9.06 | 40.69 | 8.96 | 3.72 | 48.53 | 17.72 | 0.08 | 3.12 | 21.38 |
| BGE-VL | 38.85 | 21.34 | 15.96 | 26.52 | 8.32 | 21.63 | 45.67 | 11.45 | 48.26 | 4.75 | 0.00 | 3.79 | 20.54 |
| GME | 47.02 | 46.64 | 25.82 | 27.66 | 8.40 | 36.23 | 31.39 | 4.73 | 40.24 | 12.74 | 0.10 | 7.57 | 24.04 |
| VLM2Vec | 50.97 | 49.69 | 26.93 | 31.18 | 10.83 | 43.29 | 46.70 | 10.48 | 38.68 | 9.89 | 0.62 | 3.25 | 26.88 |
| MM-Emb. | 61.30 | 59.90 | 33.43 | 41.06 | 21.60 | 54.67 | 40.32 | 19.03 | 50.30 | 18.85 | 0.09 | 3.44 | 33.67 |
| Seed-1.6 | 47.69 | 43.99 | 39.38 | 34.62 | 20.65 | 41.38 | 48.64 | 19.37 | 65.00 | 13.29 | 0.47 | 5.26 | 31.65 |

Table 14: The overall performance of embedding models on MR$^2$-Bench in terms of the MRR@10.

**Task Description:**
You are an AI assistant specializing in information retrieval and reasoning. Given an instruction and a question (consist of text and images), your task is to generate a "Chain-of-thought" reasoning process. This process must clearly outline the key information that needs to be found in relevant document to answer the question.

**Execution Flow:**
(1) Identify the Essential Problem: First, precisely extract the fundamental problem that needs to be solved.
(2) Reason on Required Information: Based on the essential problem, conduct step-by-step reasoning to specify the content that needs to be retrieved. This should include relevant terms, phenomena, causes, characteristics, risks, or solutions.
(3) Synthesize the Answer: Based on the reasoning, formulate a direct and concise answer to the problem.
(4) Combine for Output: Consolidate the "Essential Problem", the "Reasoning on Required Information", and the "Synthesized Answer" into a single, coherent text. This text must be simple, easy to understand, and kept within 100 words.

**Input Content:**
The provided instruction, question text and question images are as follows:
Original instruction: <instruction>
Original question text: <question text>
Original question images: <question images>

Figure 3: Prompt used by GPT-5 for query rewriting.

*reason-then-rank* format; the full prompt is provided in Figure 4. We evaluate Gemini-2.5-Pro[11] and GPT-5[12] using their official APIs, and BGE-Reasoner-Reranker-32B with the authors' code and checkpoint obtained via email. For open-source MLLMs (Gemma-3-27B, Qwen2.5-VL-72B, GLM-4.5V), we run inference with SGLang [13] to accelerate the reasoning stage. All other models are evaluated using their released code and checkpoints.

---

**Task Description:**

You are an objective, evidence-based multimodal judge. Given a Query and a Candidate, determine whether the Candidate appropriately corresponds to the Query (satisfies its requirements, answers its question, or retrieves the relevant information). Your task is to provide a discrete integer score from 0 to 100:

- 80-100 (Highly Relevant): The Candidate directly and comprehensively addresses the Query's intent.
- 60-80 (Relevant): The Candidate substantially addresses the Query's intent, providing most of the key information or details, but might miss some minor details.
- 40-60 (Moderately Relevant): The Candidate is relevant and addresses a part of the Query's intent, but it is not comprehensive.
- 20-40 (Slightly Relevant): The Candidate mentions some aspects about the Query, but its main intent is different. It offers very limited value or information.
- 0-20 (Irrelevant): The Candidate does not address the Query's intent at all and is off-topic or wrong.

**Reasoning Process:**

Before providing your answer, analyze the Query and the Candidate step by step and provide your analysis process:
1) **Query analysis**:
   - If the Query contains image(s): analyze the concrete visual elements (objects, attributes, colors, materials, text-in-image/OCR, spatial relations, layout/scene, etc.).
   - If the Query contains text(s): analyze the explicit intent and constraints (entities, attributes, quantities, relations, actions/edits, categories/styles, temporal/spatial cues, etc.).
   - Accurately capture the Query's true intent, identifying the key challenges and core elements.
2) **Candidate analysis**:
   - If the Candidate contains image(s): analyze the concrete visual elements (objects, attributes, colors, materials, text-in-image/OCR, spatial relations, layout/scene, etc.).
   - If the Candidate contains text: analyze its explicit content (entities, attributes, quantities, relations, categories, etc.).
   - Carefully analyze and discuss the Candidate against the Query's intent and constraints to determine whether it satisfies the Query's requirements and true intent. Avoid erroneous acceptance or rejection; base judgments strictly on observable details and reasonable reasoning.

After providing your detailed analysis and justification for all the steps above, conclude your entire response with the final score. The score must be enclosed within <score> </score> tags. Please output the score with the tag only, no other text.

Your output should follow the following format:
   your analysis process
   <score>XX</score>

---

Figure 4: Prompt used by MLLMs to score query-candidate pairs after reasoning.

## I.2  DETAILED RERANKING RESULTS WITH SEED-1.6-EMBEDDING AS THE BASE RETRIEVER

We report detailed reranking results with Seed-1.6-Embedding as the base retriever, including Recall@1, Recall@5, Recall@10, NDCG@5, NDCG@10, and NDCG@20, in Tables 15 to 20, respectively.

---

[11]gemini-2.5-pro-thinking-2025-06-05
[12]gpt-5-2025-08-07
[13]https://docs.sglang.ai/

| Methods | Multimodal Knowledge Retrieval | | | | | | Visual Illustration | | | Visual Relation | | | Avg. |
|---|---|---|---|---|---|---|---|---|---|---|---|---|---|
| | Bio. | Cook. | Gar. | Phy. | Chem. | Earth. | Econ. | Math. | Nat. | Spa. | Puzz. | Ana. | |
| *Base Retriever* | | | | | | | | | | | | | |
| Seed-1.6-Embedding | 13.65 | 9.02 | 9.85 | 5.20 | 3.69 | 9.81 | 33.33 | 6.98 | 19.33 | 8.00 | 0.00 | 0.00 | 9.91 |
| *Textual Rerankers* | | | | | | | | | | | | | |
| RankLLaMa-7B | 8.53 | 7.03 | 8.34 | 7.71 | 3.92 | 4.16 | 30.95 | 12.60 | 7.50 | 4.00 | 0.00 | 1.36 | 8.01 |
| RankLLaMa-14B | 12.00 | 8.86 | 6.42 | 4.88 | 4.12 | 7.27 | 19.05 | 13.18 | 6.25 | 7.33 | 0.62 | 0.00 | 7.50 |
| Rank1-7B | 9.33 | 6.59 | 4.32 | 3.76 | 6.14 | 6.84 | 50.00 | 31.59 | 12.25 | 8.00 | 0.00 | 0.68 | 11.63 |
| RankR1-14B | 9.86 | 4.86 | 7.00 | 6.28 | 2.88 | 7.27 | 67.86 | 17.25 | 10.58 | 13.33 | 0.63 | 2.72 | 12.54 |
| ReasonRank-32B | 13.80 | 8.96 | 8.91 | 5.46 | 4.45 | 9.78 | 58.33 | 27.03 | 10.75 | 12.00 | 0.62 | 6.80 | 13.91 |
| BGE-Reasoner-Reranker-32B | 15.43 | 6.40 | 10.29 | 4.85 | 3.97 | 11.36 | 65.48 | 25.68 | 12.00 | 18.00 | 0.62 | 2.72 | 14.73 |
| *Multimodal Rerankers* | | | | | | | | | | | | | |
| MonoQwen2-VL | 10.14 | 10.19 | 5.19 | 4.63 | 2.68 | 6.48 | 57.14 | 16.28 | 19.25 | 5.33 | 0.00 | 3.40 | 11.73 |
| Jina-Reranker | 8.18 | 7.36 | 5.53 | 3.00 | 1.68 | 5.34 | 71.43 | 24.22 | 17.00 | 25.33 | 0.00 | 2.04 | 14.26 |
| Gemma-3-27B | 9.59 | 7.15 | 3.91 | 3.52 | 6.48 | 9.94 | 36.90 | 27.81 | 11.25 | 27.33 | 1.25 | 6.80 | 12.66 |
| Qwen2.5-VL-72B | 13.21 | 10.17 | 7.39 | 5.84 | 4.48 | 11.51 | 58.33 | 40.60 | 14.58 | 28.67 | 3.12 | 5.44 | 16.95 |
| GLM-4.5V-thinking | 12.69 | 6.88 | 4.97 | 6.41 | 7.37 | 8.42 | 55.95 | 39.15 | 12.42 | 32.00 | 1.88 | 5.44 | 16.13 |
| Gemini-2.5-Pro | 9.84 | 13.64 | 9.20 | 7.35 | 9.31 | 11.16 | 58.33 | 42.34 | 23.58 | 40.67 | 0.00 | 10.20 | 19.64 |
| GPT-5 | 16.66 | 17.05 | 12.37 | 8.21 | 11.29 | 16.39 | 77.38 | 50.48 | 26.17 | 39.33 | 2.50 | 11.56 | 24.12 |

Table 15: Detailed reranking performance (Recall@1) on MR$^2$-Bench with Seed-1.6-Embedding as the base retriever.

| Methods | Multimodal Knowledge Retrieval | | | | | | Visual Illustration | | | Visual Relation | | | Avg. |
|---|---|---|---|---|---|---|---|---|---|---|---|---|---|
| | Bio. | Cook. | Gar. | Phy. | Chem. | Earth. | Econ. | Math. | Nat. | Spa. | Puzz. | Ana. | |
| *Base Retriever* | | | | | | | | | | | | | |
| Seed-1.6-Embedding | 31.93 | 32.51 | 28.95 | 22.17 | 14.52 | 31.65 | 69.05 | 38.76 | 61.25 | 19.33 | 0.63 | 8.16 | 29.91 |
| *Textual Rerankers* | | | | | | | | | | | | | |
| RankLLaMa-7B | 32.34 | 30.02 | 26.59 | 21.19 | 11.61 | 25.54 | 71.43 | 42.73 | 34.42 | 20.00 | 1.25 | 8.16 | 27.11 |
| RankLLaMa-14B | 33.86 | 29.62 | 26.57 | 21.86 | 16.97 | 33.26 | 64.29 | 47.38 | 28.42 | 18.67 | 1.25 | 6.80 | 27.41 |
| Rank1-7B | 30.62 | 31.07 | 22.42 | 16.12 | 17.16 | 25.23 | 77.38 | 52.03 | 40.50 | 22.67 | 1.88 | 6.80 | 28.66 |
| RankR1-14B | 32.09 | 32.21 | 23.81 | 19.68 | 18.97 | 27.78 | 84.52 | 52.03 | 37.50 | 32.67 | 2.50 | 17.01 | 31.73 |
| ReasonRank-32B | 33.17 | 30.95 | 26.29 | 20.90 | 15.97 | 29.08 | 80.95 | 52.62 | 43.83 | 20.67 | 3.12 | 19.73 | 31.44 |
| BGE-Reasoner-Reranker-32B | 36.79 | 35.21 | 28.52 | 18.56 | 16.75 | 29.96 | 83.33 | 53.78 | 47.83 | 33.33 | 3.12 | 12.24 | 33.29 |
| *Multimodal Rerankers* | | | | | | | | | | | | | |
| MonoQwen2-VL | 27.49 | 35.15 | 23.82 | 15.77 | 12.98 | 22.15 | 79.76 | 51.45 | 60.58 | 18.67 | 0.62 | 14.29 | 30.23 |
| Jina-Reranker | 27.42 | 30.17 | 25.63 | 16.59 | 14.57 | 20.78 | 85.71 | 53.20 | 60.92 | 35.33 | 0.00 | 12.93 | 31.94 |
| Gemma-3-27B | 34.17 | 35.13 | 25.76 | 17.03 | 24.54 | 26.95 | 70.24 | 53.20 | 52.50 | 36.67 | 2.50 | 19.73 | 33.20 |
| Qwen2.5-VL-72B | 33.76 | 29.68 | 27.45 | 18.45 | 19.08 | 31.07 | 83.33 | 54.94 | 59.33 | 32.67 | 5.62 | 14.29 | 34.14 |
| GLM-4.5V-thinking | 33.73 | 39.96 | 24.50 | 19.31 | 20.34 | 30.69 | 80.95 | 56.10 | 60.58 | 38.00 | 5.00 | 24.49 | 36.14 |
| Gemini-2.5-Pro | 40.03 | 43.72 | 31.12 | 21.79 | 24.69 | 38.95 | 82.14 | 53.78 | 76.17 | 42.67 | 3.28 | 26.53 | 40.41 |
| GPT-5 | 46.39 | 51.27 | 33.65 | 27.32 | 28.86 | 48.46 | 88.10 | 56.10 | 79.33 | 43.33 | 4.38 | 27.21 | 44.53 |

Table 16: Detailed reranking performance (Recall@5) on MR$^2$-Bench with Seed-1.6-Embedding as the base retriever.

| Methods | Multimodal Knowledge Retrieval | | | | | | Visual Illustration | | | Visual Relation | | | Avg. |
|---|---|---|---|---|---|---|---|---|---|---|---|---|---|
| | Bio. | Cook. | Gar. | Phy. | Chem. | Earth. | Econ. | Math. | Nat. | Spa. | Puzz. | Ana. | |
| *Base Retriever* | | | | | | | | | | | | | |
| Seed-1.6-Embedding | 47.99 | 49.13 | 38.60 | 30.05 | 26.32 | 48.90 | 79.76 | 47.87 | 84.17 | 30.67 | 2.50 | 22.45 | 42.37 |
| *Textual Rerankers* | | | | | | | | | | | | | |
| RankLLaMa-7B | 47.24 | 48.26 | 39.69 | 30.41 | 23.05 | 46.68 | 82.14 | 51.45 | 63.17 | 30.00 | 3.12 | 13.61 | 39.90 |
| RankLLaMa-14B | 51.11 | 53.74 | 39.11 | 31.22 | 28.30 | 54.12 | 78.57 | 51.45 | 56.67 | 30.67 | 3.12 | 19.05 | 41.43 |
| Rank1-7B | 45.92 | 48.38 | 35.68 | 29.07 | 28.97 | 48.48 | 85.71 | 54.94 | 65.58 | 30.00 | 5.00 | 14.97 | 41.06 |
| RankR1-14B | 49.81 | 50.16 | 36.98 | 30.27 | 28.91 | 48.04 | 86.90 | 56.10 | 67.92 | 36.67 | 3.12 | 26.53 | 43.45 |
| ReasonRank-32B | 47.86 | 47.93 | 36.96 | 30.54 | 24.67 | 45.45 | 85.71 | 53.78 | 62.92 | 33.33 | 4.38 | 25.85 | 41.61 |
| BGE-Reasoner-Reranker-32B | 50.54 | 52.14 | 41.71 | 30.82 | 30.02 | 51.88 | 85.71 | 56.10 | 75.00 | 41.33 | 4.38 | 23.81 | 45.29 |
| *Multimodal Rerankers* | | | | | | | | | | | | | |
| MonoQwen2-VL | 40.82 | 45.84 | 33.23 | 26.22 | 24.64 | 40.45 | 83.33 | 56.10 | 86.83 | 31.33 | 3.12 | 25.17 | 41.43 |
| Jina-Reranker | 42.93 | 47.88 | 38.99 | 29.59 | 26.99 | 37.69 | 88.10 | 55.52 | 85.25 | 40.00 | 1.88 | 24.49 | 43.28 |
| Gemma-3-27B | 49.26 | 55.19 | 36.78 | 28.35 | 33.09 | 45.52 | 84.52 | 54.94 | 79.92 | 44.00 | 3.12 | 29.25 | 45.33 |
| Qwen2.5-VL-72B | 50.40 | 51.51 | 41.59 | 28.24 | 30.37 | 49.01 | 85.71 | 56.10 | 86.58 | 36.67 | 5.62 | 27.21 | 45.75 |
| GLM-4.5V-thinking | 50.55 | 54.87 | 38.50 | 31.67 | 30.79 | 46.73 | 84.52 | 56.10 | 85.67 | 39.33 | 5.62 | 30.61 | 46.25 |
| Gemini-2.5-Pro | 54.16 | 58.46 | 42.45 | 32.32 | 36.61 | 58.23 | 88.10 | 53.78 | 94.83 | 44.00 | 6.56 | 29.93 | 49.95 |
| GPT-5 | 56.56 | 60.34 | 44.35 | 38.33 | 37.87 | 60.88 | 88.10 | 56.10 | 94.67 | 44.00 | 5.62 | 31.29 | 51.51 |

Table 17: Detailed reranking performance (Recall@10) on MR$^2$-Bench with Seed-1.6-Embedding as the base retriever.

| Methods | Multimodal Knowledge Retrieval | | | | | | Visual Illustration | | | Visual Relation | | | Avg. |
|---|---|---|---|---|---|---|---|---|---|---|---|---|---|
| | Bio. | Cook. | Gar. | Phy. | Chem. | Earth. | Econ. | Math. | Nat. | Spa. | Puzz. | Ana. | |
| *Base Retriever* | | | | | | | | | | | | | |
| Seed-1.6-Embedding | 36.14 | 32.45 | 28.34 | 27.69 | 13.46 | 31.52 | 52.63 | 22.95 | 55.12 | 13.67 | 0.31 | 4.49 | 26.56 |
| *Textual Rerankers* | | | | | | | | | | | | | |
| RankLLaMa-7B | 33.57 | 27.86 | 25.75 | 25.33 | 12.71 | 22.03 | 51.12 | 29.83 | 28.23 | 11.95 | 0.48 | 4.72 | 22.80 |
| RankLLaMa-14B | 37.46 | 30.23 | 25.21 | 24.77 | 16.90 | 29.61 | 41.34 | 33.38 | 23.23 | 12.70 | 0.89 | 3.42 | 23.26 |
| Rank1-7B | 32.41 | 28.02 | 19.18 | 22.60 | 17.04 | 24.76 | 64.57 | 44.02 | 36.11 | 15.45 | 0.91 | 3.70 | 25.73 |
| RankR1-14B | 35.23 | 27.72 | 21.90 | 27.27 | 15.87 | 26.91 | 76.71 | 38.19 | 32.75 | 23.10 | 1.73 | 10.14 | 28.13 |
| ReasonRank-32B | 38.20 | 31.44 | 24.65 | 28.31 | 15.76 | 30.00 | 70.57 | 43.11 | 36.98 | 16.85 | 1.87 | 13.20 | 29.25 |
| BGE-Reasoner-Reranker-32B | 41.86 | 33.58 | 26.89 | 27.38 | 16.31 | 31.51 | 75.80 | 43.57 | 41.68 | 25.80 | 1.72 | 7.53 | 31.14 |
| *Multimodal Rerankers* | | | | | | | | | | | | | |
| MonoQwen2-VL | 31.17 | 32.96 | 21.96 | 22.14 | 11.02 | 21.24 | 69.47 | 37.18 | 53.35 | 12.07 | 0.39 | 8.84 | 26.82 |
| Jina-Reranker | 28.54 | 29.05 | 23.35 | 20.17 | 11.03 | 20.04 | 78.68 | 42.30 | 52.74 | 30.42 | 0.00 | 7.61 | 28.66 |
| Gemma-3-27B | 36.21 | 31.97 | 21.52 | 22.28 | 22.53 | 27.05 | 55.09 | 43.53 | 42.35 | 32.61 | 1.73 | 13.40 | 29.19 |
| Qwen2.5-VL-72B | 38.51 | 30.82 | 23.70 | 27.05 | 17.25 | 30.75 | 71.80 | 50.71 | 49.40 | 30.73 | 4.58 | 10.21 | 32.13 |
| GLM-4.5V-thinking | 37.66 | 35.63 | 20.59 | 27.83 | 21.15 | 28.66 | 69.38 | 50.24 | 49.37 | 35.61 | 3.60 | 15.25 | 32.92 |
| Gemini-2.5-Pro | 42.94 | 45.34 | 28.56 | 29.58 | 24.80 | 37.80 | 71.09 | 50.94 | 67.46 | 41.59 | 1.74 | 18.46 | 38.36 |
| GPT-5 | 52.03 | 54.15 | 34.19 | 35.91 | 29.34 | 48.00 | 83.83 | 55.63 | 72.09 | 41.21 | 3.53 | 20.19 | 44.18 |

Table 18: Detailed reranking performance (NDCG@5) on MR$^2$-Bench with Seed-1.6-Embedding as the base retriever.

| Methods | Multimodal Knowledge Retrieval | | | | | | Visual Illustration | | | Visual Relation | | | Avg. |
|---|---|---|---|---|---|---|---|---|---|---|---|---|---|
| | Bio. | Cook. | Gar. | Phy. | Chem. | Earth. | Econ. | Math. | Nat. | Spa. | Puzz. | Ana. | |
| *Base Retriever* | | | | | | | | | | | | | |
| Seed-1.6-Embedding | 40.64 | 38.12 | 31.77 | 27.91 | 17.80 | 37.17 | 56.13 | 26.10 | 65.16 | 17.29 | 0.93 | 9.21 | 30.68 |
| *Textual Rerankers* | | | | | | | | | | | | | |
| RankLLaMa-7B | 37.92 | 34.53 | 30.40 | 27.74 | 16.31 | 30.35 | 54.62 | 32.71 | 40.76 | 15.21 | 1.04 | 6.42 | 27.33 |
| RankLLaMa-14B | 43.27 | 38.94 | 29.61 | 26.92 | 20.32 | 37.03 | 45.98 | 34.65 | 35.08 | 16.62 | 1.49 | 7.20 | 28.09 |
| Rank1-7B | 36.95 | 35.07 | 24.49 | 25.34 | 21.21 | 33.87 | 67.19 | 45.01 | 47.22 | 17.39 | 1.96 | 6.29 | 30.21 |
| RankR1-14B | 40.11 | 34.96 | 26.88 | 28.58 | 19.60 | 34.36 | 77.49 | 39.56 | 46.30 | 24.41 | 1.93 | 13.19 | 32.28 |
| ReasonRank-32B | 41.70 | 37.31 | 28.98 | 28.34 | 18.61 | 35.57 | 72.19 | 43.53 | 45.35 | 21.10 | 2.30 | 15.16 | 32.51 |
| BGE-Reasoner-Reranker-32B | 45.19 | 39.31 | 32.18 | 28.57 | 20.69 | 39.26 | 76.53 | 44.35 | 53.07 | 28.35 | 2.09 | 11.31 | 35.08 |
| *Multimodal Rerankers* | | | | | | | | | | | | | |
| MonoQwen2-VL | 35.33 | 36.41 | 24.71 | 23.96 | 15.31 | 27.96 | 70.60 | 38.93 | 64.83 | 18.23 | 1.14 | 12.28 | 30.64 |
| Jina-Reranker | 34.23 | 35.45 | 28.13 | 24.25 | 15.67 | 25.86 | 79.48 | 43.21 | 63.63 | 31.90 | 0.60 | 11.35 | 32.82 |
| Gemma-3-27B | 39.94 | 39.67 | 26.15 | 25.57 | 25.11 | 33.94 | 59.75 | 44.20 | 54.44 | 34.99 | 1.96 | 16.44 | 33.51 |
| Qwen2.5-VL-72B | 42.95 | 38.78 | 29.60 | 28.21 | 21.17 | 37.66 | 72.57 | 51.09 | 61.47 | 31.97 | 4.58 | 14.29 | 36.20 |
| GLM-4.5V-thinking | 42.43 | 41.28 | 26.37 | 29.78 | 24.34 | 34.15 | 70.52 | 50.24 | 60.24 | 36.06 | 3.78 | 17.29 | 36.37 |
| Gemini-2.5-Pro | 45.91 | 49.72 | 33.10 | 30.87 | 28.57 | 44.28 | 73.13 | 50.94 | 76.07 | 42.02 | 2.87 | 19.61 | 41.43 |
| GPT-5 | 52.35 | 55.41 | 37.46 | 37.10 | 31.96 | 51.12 | 83.83 | 55.63 | 79.16 | 41.41 | 3.94 | 21.48 | 45.90 |

Table 19: Detailed reranking performance (nDCG@10) on MR$^2$-Bench with Seed-1.6-Embedding as the base retriever.

| Methods | Multimodal Knowledge Retrieval | | | | | | Visual Illustration | | | Visual Relation | | | Avg. |
|---|---|---|---|---|---|---|---|---|---|---|---|---|---|
| | Bio. | Cook. | Gar. | Phy. | Chem. | Earth. | Econ. | Math. | Nat. | Spa. | Puzz. | Ana. | |
| *Base Retriever* | | | | | | | | | | | | | |
| Seed-1.6-Embedding | 46.01 | 43.31 | 35.86 | 32.99 | 22.85 | 43.71 | 58.25 | 28.38 | 69.97 | 21.20 | 1.67 | 11.76 | 34.66 |
| *Textual Rerankers* | | | | | | | | | | | | | |
| RankLLaMa-7B | 43.30 | 39.99 | 34.24 | 33.08 | 22.25 | 37.77 | 56.11 | 34.04 | 52.68 | 19.21 | 1.66 | 11.14 | 32.12 |
| RankLLaMa-14B | 46.82 | 42.36 | 33.27 | 31.74 | 24.48 | 42.40 | 48.40 | 35.90 | 49.12 | 20.54 | 2.13 | 10.52 | 32.31 |
| Rank1-7B | 43.28 | 41.15 | 30.00 | 30.70 | 25.27 | 40.54 | 67.78 | 45.31 | 57.94 | 21.86 | 2.12 | 10.74 | 34.72 |
| RankR1-14B | 44.82 | 40.37 | 31.75 | 33.63 | 23.64 | 41.27 | 77.81 | 39.56 | 56.77 | 26.81 | 2.56 | 14.77 | 36.15 |
| ReasonRank-32B | 46.94 | 42.68 | 33.66 | 33.36 | 24.11 | 43.35 | 72.81 | 44.07 | 57.47 | 24.32 | 2.62 | 16.88 | 36.86 |
| BGE-Reasoner-Reranker-32B | 49.23 | 43.49 | 35.21 | 32.73 | 24.16 | 44.67 | 77.16 | 44.35 | 60.96 | 29.55 | 2.43 | 13.62 | 38.13 |
| *Multimodal Rerankers* | | | | | | | | | | | | | |
| MonoQwen2-VL | 43.11 | 43.47 | 30.86 | 30.55 | 21.21 | 37.72 | 71.78 | 38.93 | 68.60 | 19.96 | 1.79 | 14.25 | 35.19 |
| Jina-Reranker | 41.56 | 41.24 | 32.18 | 29.78 | 20.68 | 36.60 | 79.48 | 43.40 | 67.91 | 33.44 | 1.56 | 13.44 | 36.77 |
| Gemma-3-27B | 44.65 | 42.71 | 31.30 | 30.73 | 27.69 | 41.96 | 60.70 | 44.51 | 60.61 | 35.52 | 2.58 | 17.30 | 36.69 |
| Qwen2.5-VL-72B | 47.32 | 43.21 | 33.16 | 33.36 | 24.65 | 43.95 | 73.22 | 51.09 | 65.48 | 34.26 | 4.58 | 15.65 | 39.16 |
| GLM-4.5V-thinking | 46.89 | 44.25 | 30.64 | 33.77 | 27.29 | 41.36 | 71.39 | 50.24 | 64.66 | 37.68 | 3.78 | 17.77 | 39.14 |
| Gemini-2.5-Pro | 47.75 | 50.48 | 35.75 | 34.10 | 29.81 | 47.64 | 73.13 | 51.20 | 77.27 | 42.50 | 2.87 | 20.34 | 42.74 |
| GPT-5 | 53.91 | 55.97 | 39.24 | 38.50 | 32.61 | 53.21 | 83.83 | 55.63 | 80.30 | 41.95 | 3.94 | 21.84 | 46.75 |

Table 20: Detailed reranking performance (NDCG@20) on MR$^2$-Bench with Seed-1.6-Embedding as the base retriever.

## I.3    ADDITIONAL RERANKING RESULTS FOR OTHER RETRIEVERS

We report NDCG@10 reranking results for two additional retrievers, Qwen3-Embedding and GME, in Tables 21 and 22, respectively.

| Methods | Multimodal Knowledge Retrieval | | | | | | Visual Illustration | | | Visual Relation | | | Avg. |
|---|---|---|---|---|---|---|---|---|---|---|---|---|---|
| | Bio. | Cook. | Gar. | Phy. | Chem. | Earth. | Econ. | Math. | Nat. | Spa. | Puzz. | Ana. | |
| *Base Retriever* | | | | | | | | | | | | | |
| Qwen3-Embedding | 29.97 | 29.29 | 18.32 | 21.46 | 9.52 | 23.19 | 49.44 | 21.14 | 26.30 | 9.11 | 0.00 | 4.30 | 20.17 |
| *Textual Rerankers* | | | | | | | | | | | | | |
| RankLLaMa-7B | 32.85 | 30.80 | 22.53 | 25.89 | 12.95 | 27.73 | 47.93 | 31.12 | 26.69 | 9.19 | 0.00 | 5.96 | 22.80 |
| RankLLaMa-14B | 39.06 | 33.81 | 22.29 | 27.29 | 17.24 | 32.90 | 40.37 | 31.21 | 21.41 | 10.43 | 0.00 | 4.92 | 23.41 |
| Rank1-7B | 33.62 | 30.46 | 19.22 | 20.56 | 13.44 | 26.24 | 51.16 | 38.62 | 32.60 | 12.38 | 0.00 | 3.72 | 23.50 |
| RankR1-14B | 37.54 | 31.86 | 22.30 | 25.67 | 14.26 | 27.49 | 59.58 | 35.66 | 31.97 | 14.25 | 0.00 | 6.61 | 25.60 |
| ReasonRank-32B | 36.24 | 30.69 | 20.63 | 25.53 | 13.09 | 26.45 | 60.59 | 36.26 | 34.46 | 12.16 | 0.00 | 7.59 | 25.31 |
| BGE-Reasoner-Reranker-32B | 39.97 | 33.44 | 22.20 | 23.68 | 14.12 | 29.83 | 62.49 | 41.27 | 32.14 | 15.66 | 0.00 | 7.34 | 26.85 |
| *Multimodal Rerankers* | | | | | | | | | | | | | |
| MonoQwen2-VL | 32.13 | 32.15 | 20.53 | 22.02 | 11.72 | 22.22 | 58.21 | 33.62 | 41.02 | 10.17 | 0.00 | 7.57 | 24.28 |
| Jina-Reranker | 32.22 | 30.26 | 21.41 | 22.24 | 10.46 | 22.55 | 63.47 | 38.04 | 41.69 | 18.42 | 0.00 | 6.49 | 25.60 |
| Gemma-3-27B | 35.20 | 33.70 | 19.85 | 22.10 | 16.81 | 26.66 | 48.22 | 39.52 | 35.59 | 18.79 | 0.00 | 7.02 | 25.29 |
| Qwen2.5-VL-72B | 40.11 | 36.20 | 20.69 | 24.94 | 14.79 | 28.74 | 58.25 | 46.29 | 38.55 | 17.92 | 0.00 | 6.92 | 27.78 |
| GLM-4.5V | 36.09 | 34.75 | 18.83 | 25.53 | 16.45 | 27.06 | 55.29 | 42.19 | 39.76 | 18.93 | 0.00 | 7.92 | 26.90 |

Table 21: Detailed reranking performance (nDCG@10) on MR$^2$-Bench with Qwen3-Embedding as the base retriever.

| Methods | Multimodal Knowledge Retrieval | | | | | | Visual Illustration | | | Visual Relation | | | Avg. |
|---|---|---|---|---|---|---|---|---|---|---|---|---|---|
| | Bio. | Cook. | Gar. | Phy. | Chem. | Earth. | Econ. | Math. | Nat. | Spa. | Puzz. | Ana. | |
| *Base Retriever* | | | | | | | | | | | | | |
| GME | 34.34 | 39.50 | 19.04 | 19.29 | 7.73 | 28.59 | 36.95 | 7.19 | 39.35 | 15.70 | 0.22 | 11.11 | 21.59 |
| *Textual Rerankers* | | | | | | | | | | | | | |
| RankLLaMa-7B | 30.58 | 28.35 | 14.86 | 23.71 | 10.44 | 26.29 | 48.66 | 13.48 | 34.31 | 11.03 | 0.00 | 9.39 | 20.92 |
| RankLLaMa-14B | 33.92 | 32.62 | 13.82 | 22.35 | 11.98 | 32.82 | 40.31 | 15.34 | 29.22 | 15.76 | 0.00 | 8.57 | 21.39 |
| Rank1-7B | 32.05 | 37.15 | 17.50 | 20.11 | 12.04 | 30.06 | 55.96 | 19.14 | 39.81 | 16.41 | 0.00 | 6.87 | 23.92 |
| RankR1-14B | 35.39 | 35.87 | 19.49 | 23.89 | 12.38 | 29.86 | 59.66 | 16.84 | 40.43 | 20.26 | 0.00 | 10.54 | 25.38 |
| ReasonRank-32B | 34.49 | 36.50 | 20.02 | 23.49 | 12.45 | 30.21 | 59.08 | 18.86 | 37.60 | 17.25 | 0.00 | 11.32 | 25.11 |
| BGE-Reasoner-Reranker-32B | 37.05 | 40.29 | 21.98 | 22.33 | 14.24 | 32.43 | 62.27 | 18.45 | 44.00 | 24.13 | 0.00 | 11.24 | 27.37 |
| *Multimodal Rerankers* | | | | | | | | | | | | | |
| MonoQwen2-VL | 31.61 | 35.58 | 17.13 | 19.82 | 9.49 | 23.25 | 60.35 | 14.81 | 55.25 | 13.87 | 0.24 | 11.42 | 24.40 |
| Jina-Reranker | 31.97 | 36.23 | 19.39 | 19.56 | 8.92 | 23.78 | 63.58 | 17.20 | 54.47 | 23.24 | 0.39 | 10.29 | 25.84 |
| Gemma-3-27B | 36.20 | 42.07 | 19.72 | 21.16 | 14.82 | 27.93 | 49.19 | 19.01 | 44.77 | 26.94 | 0.00 | 16.21 | 26.50 |
| Qwen2.5-VL-72B | 35.36 | 38.93 | 21.18 | 20.97 | 13.56 | 30.49 | 57.23 | 21.41 | 49.84 | 26.62 | 0.20 | 16.37 | 27.68 |
| GLM-4.5V | 35.60 | 41.26 | 18.95 | 21.10 | 14.53 | 28.70 | 55.39 | 20.21 | 52.55 | 30.73 | 0.62 | 17.83 | 28.12 |

Table 22: Detailed reranking performance (nDCG@10) on MR$^2$-Bench with GME as the base retriever.

