# OpenReview forum: "MR$^2$-Bench: Going Beyond Matching to Reasoning in Multimodal Retrieval"
_ICLR.cc/2026/Conference — Submitted to ICLR 2026_

### Official Review · Reviewer_e2GU · 2025-10-25

**Soundness:** 3
**Presentation:** 3
**Contribution:** 2
**Rating:** 6
**Confidence:** 4

**Summary:**

The paper proposes a new benchmark, MR2-Bench, for "reasoning-intensive multimodal retrieval." The benchmark covers three meta-tasks (multimodal knowledge retrieval, visual illustration search, and visual relation reasoning) and twelve sub-tasks, comprising a total of 1,309 queries. It emphasizes flexible interleaved text-image formats, single/multi-image inputs, multi-domain coverage, and evaluation criteria requiring logical, spatial, and causal reasoning (Table 1 and Table 2). The authors systematically evaluate eleven text and multimodal embedding retrieval models using nDCG@10 as the primary metric, concluding that current SOTA methods fail significantly on MR2-Bench. For example, Seed-1.6-Embedding achieves Recall@1 = 77.78 on MMEB but only 9.91 on MR2-Bench, with an average nDCG@10 of 30.68. The authors further present two enhancement techniques: retrieval-oriented Query Rewriting (using GPT-5 to generate step-by-step reasoning and reformulate queries) and Reranking (using explicit multimodal re-rankers that “reason before scoring”), both yielding substantial improvements in average performance (Table 4, Figure 2).

**Strengths:**

1. Clearly defines “reasoning” as the prerequisite for relevance judgment, distinguishing MR2-Bench from prior benchmarks focused on shallow matching. It introduces data types rarely included in retrieval evaluation before, such as visual puzzles, mathematical visual proofs, and economic charts, making the task design both novel and necessary.

2. The benchmark allows both queries and candidates to include multiple images or interleaved text-image formats, reflecting real-world document retrieval structures rarely seen in existing benchmarks.

3. Task and data statistics are clearly presented. The three meta-tasks and twelve sub-tasks specify query and corpus scales (Table 2), covering diverse scenarios from natural science to visual reasoning.

4. The result analysis captures the essence of “why it is difficult”: multi-image relations, spatial reasoning, and abstract figures represent systematic weaknesses of current methods.

5. The main findings are straightforward: SOTA models significantly degrade on MR2-Bench; caption enhancement notably improves text retrievers; and introducing “reason-before-ranking” strategies is effective.

6. The quantified performance gap (e.g., Seed-1.6 Recall@1 comparison between MMEB and MR2-Bench) strongly supports the claim that reasoning-based retrieval is more challenging and valuable for research than prior evaluations.

**Weaknesses:**

1. The positive document selection and hard negative sampling in the multimodal knowledge retrieval subset rely on GPT-5 for initial filtering, followed by expert review. Some hard negatives are GPT-5-generated paragraphs that are “topic-relevant but unhelpful.” Although human validation is included, this pipeline introduces unquantified model bias and potential data contamination risks (e.g., “tool consistency bias” with GPT-5 also used in the reranking/reformulation stage). The paper should provide stricter inter-annotator agreement metrics and cross-model sensitivity analyses.

2. No statistics are provided on annotation consistency (e.g., IAA/Kappa) or multi-round review ratios, making it difficult to assess the stability and transferability of labeling quality.

3. The text retrievers replace all images with captions generated by the same model (Qwen2.5-VL-7B) using fixed prompts, which may induce systematic bias favoring certain methods (e.g., those aligned with the captioning style). The paper does not report sensitivity to different caption models or prompts, nor does it compare human-written summaries versus automatic captions.

4. The total of 1,309 queries is relatively small compared with general benchmarks such as M-BEIR (190,000), and sub-task corpus sizes vary widely (e.g., Mathematics: 944 vs. Economics: 7,572), potentially introducing uneven weighting in averaged metrics. Weighted indicators or significance tests across tasks are recommended.

5. Although the benchmark emphasizes “free-form documents and interleaved multi-image inputs,” evaluation remains limited to offline retrieval metrics (nDCG/Recall). Adding out-of-distribution queries, cross-domain transfer (e.g., new chart sources or species), and analysis linking retrieval quality with end-to-end RAG answer accuracy would strengthen claims of real-world applicability.

6. External data sources such as economic charts (World Bank reports), natural images (INQUIRE-Rerank), and datasets like RAVEN/CSS/VASR are mentioned in the main text/appendix, but there is no consolidated ethical section detailing licensing, re-distribution permissions, or potential copyright/portrait issues. The paper should include a dedicated ethics subsection enumerating data sources, licenses, usage boundaries, and de-identification procedures.

**Questions:**

Questions correspond to the issues listed in the Weaknesses section.

**Details Of Ethics Concerns:**

The paper uses charts from World Bank reports, natural images from INQUIRE-Rerank, and public datasets such as RAVEN, CSS, and VASR, but neither the main text nor the appendix provides a consolidated presentation of copyright, licensing, or re-distribution terms for each source. To ensure reproducibility and compliance, it is recommended to add an “Ethics and Data Licensing” section that explicitly lists each source, license type, whether redistribution is allowed, and whether attribution is required or citation-only use is permitted.

---

> ### Author Response · Authors · 2025-11-21
>
> Dear Reviewer,
>
> We are grateful for your positive evaluation of our work and your valuable suggestions. Please find our responses below.
>
> > The positive document selection and hard negative sampling in the multimodal knowledge retrieval subset rely on GPT-5 for initial filtering, followed by expert review. Some hard negatives are GPT-5-generated paragraphs that are “topic-relevant but unhelpful.” Although human validation is included, this pipeline introduces unquantified model bias and potential data contamination risks (e.g., “tool consistency bias” with GPT-5 also used in the reranking/reformulation stage). The paper should provide stricter inter-annotator agreement metrics and cross-model sensitivity analyses.
>
> Thank you for your insightful feedback! We appreciate the opportunity to address the concerns raised regarding the potential biases introduced by the use of GPT-5.
>
> To ensure that the observed performance gains are not solely attributable to "tool consistency bias" (i.e., overfitting to GPT-5's preferences), we conducted an evaluation using **Gemini-2.5-Pro**, an independent state-of-the-art model that was not involved in the data construction process. Gemini-2.5-Pro achieved an nDCG@10 score of **41.43**, significantly outperforming the base retriever (30.68) and approaching the performance of GPT-5 (45.90). While it trails slightly behind GPT-5, this performance margin mirrors the landscape observed in other established reasoning-intensive benchmarks. For instance, on MMMU [1], GPT-5 (84.0) similarly maintains a lead over Gemini-2.5-Pro (80.0). A consistent gap is also evident on the highly challenging HLE (Humanity's Last Exam) benchmark [2], where GPT-5 achieves 25.0 compared to Gemini-2.5-Pro's 18.0. This cross-benchmark consistency strongly suggests that the score difference on MR$^2$-Bench reflects genuine differences in model reasoning capabilities rather than an artifact of data construction bias. These results have been updated in Figure 2 of the paper.
>
> We also want to clarify that GPT-5 was used solely as a proposal tool for generating potential hard negatives. Crucially, all positive documents and generated hard negatives underwent human validation. To quantitatively assess the reliability of this human review process and ensure it effectively mitigates potential model bias, we conducted a rigorous Annotation Consistency Study. We achieved a Cohen’s Kappa ($\kappa$) of 0.85, which indicates "almost perfect" agreement according to standard benchmarks [3]. This high agreement confirms that our human annotators maintained consistent and independent judgment, regardless of the proposal source. Please refer to our response to the next question (regarding annotation consistency) for the detailed methodology and statistics.
>
> [1] Yue, Xiang, et al. "Mmmu: A massive multi-discipline multimodal understanding and reasoning benchmark for expert agi." Proceedings of the IEEE/CVF Conference on Computer Vision and Pattern Recognition. 2024.
>
> [2] Phan, Long, et al. "Humanity's last exam." arXiv preprint arXiv:2501.14249 (2025).
>
> [3] Landis, J. Richard, and Gary G. Koch. "The measurement of observer agreement for categorical data." biometrics (1977): 159-174.

---

> ### Author Response · Authors · 2025-11-21
>
> > No statistics are provided on annotation consistency (e.g., IAA/Kappa) or multi-round review ratios, making it difficult to assess the stability and transferability of labeling quality.
>
> Thank you for your insightful feedback! We agree that quantitative agreement metrics are essential for assessing the stability and transferability of labeling quality, and we appreciate your suggestion to provide such statistics.
>
> In response to your comments, we conducted an Inter-Annotator Agreement (IAA) study to evaluate the consistency of our annotations. To ensure a comprehensive evaluation, we adopted a stratified sampling strategy: we independently randomly sampled 200 queries to validate positive pairs and another 200 queries to validate negative pairs (covering both GPT-5 generated hard negatives and hard negatives retrieved from our internal corpus).
>
> We employed a pair-wise validation approach where two annotators independently classified the document as either a positive or negative example given the original query. The cross-validation results are presented in the confusion matrix below:
>
> | Annotator 1 \ Annotator 2 | Positive | Negative | **Total** |
> | :--- | :---: | :---: | :---: |
> | **Positive** | **653** | 86 | 739 |
> | **Negative** | 50 | **1092** | 1142 |
> | **Total** | 703 | 1178 | **1881** |
>
> *Note: The numbers in the table represent query-document pairs. Since MR$^2$-Bench queries are associated with multiple positive and hard negative candidates, the total number of validated pairs (1,881) exceeds the number of sampled queries.*
>
> Based on these results, we calculated the Inter-Annotator Agreement (IAA) as 92.8% and Cohen’s Kappa ($\kappa$) as 0.85. According to standard benchmarks [1], a Kappa value above 0.81 indicates "almost perfect" agreement.
>
> These high metrics demonstrate the consistency of our annotation process. We are confident that this additional validation confirms the robustness of our labeling pipeline and the quality of the dataset. Thank you again for helping us strengthen our paper.
>
> [1] Landis, J. Richard, and Gary G. Koch. "The measurement of observer agreement for categorical data." biometrics (1977): 159-174.

---

> ### Author Response · Authors · 2025-11-21
>
> > The text retrievers replace all images with captions generated by the same model (Qwen2.5-VL-7B) using fixed prompts, which may induce systematic bias favoring certain methods (e.g., those aligned with the captioning style). The paper does not report sensitivity to different caption models or prompts, nor does it compare human-written summaries versus automatic captions.
>
> Thank you for your valuable feedback and insightful comments. We truly appreciate your suggestion regarding the potential systematic bias introduced by the use of a fixed captioning model (Qwen2.5-VL-7B) in our experiments, and we would like to address your concerns.
>
> To clarify, the captions generated by Qwen2.5-VL-7B are not considered part of the dataset itself, but rather serve as a method to implement multimodal retrieval. These captions were generated to augment the text retrievers with additional context, facilitating the retrieval of relevant information from both text and image data. We do not view the use of captions as introducing bias into the dataset, but rather as a tool that helps bridge the gap between textual and visual data in multimodal tasks. We believe this approach is consistent with the goal of enhancing the retriever's ability to process both modalities in a unified manner.
>
> To address your concern about the potential bias of the captioning model, we conducted a sensitivity analysis to verify whether the performance of text-based retrievers is influenced by the choice of captioning model. In this analysis, we replaced the original captions with those generated by **GLM-4.1V-9B-Thinking** ([link to model](https://huggingface.co/zai-org/GLM-4.1V-9B-Thinking)), a state-of-the-art multimodal language model known for its chain-of-thought reasoning capabilities. The results show that, despite some fluctuations in absolute performance across different sub-tasks, the **relative ranking** of the retrievers remained highly consistent. Specifically, reasoning-enhanced retrievers (**ReasonIR** and **Diver-Retriever**) consistently outperformed the standard retrievers, regardless of whether the captions were generated by Qwen2.5-VL-7B** or **GLM-4.1V-9B-Thinking.
>
> | Methods            | Qwen2.5-VL-7B (Captions) | GLM-4.1V-9B-Thinking (Captions) |
> |--------------------|--------------------------|--------------------------------|
> | **ReasonIR**        | 25.72                    | 26.26                          |
> | **Diver-Retriever** | 23.59                    | 24.38                          |
> | **Qwen3-Embedding** | 20.17                    | 19.32                          |
> | **BGE-M3**          | 18.71                    | 18.59                          |
>
> We have included a detailed discussion and the full analysis in **Appendix F** and **Table 7** in the revised manuscript, which provides a comprehensive comparison across all 12 sub-tasks. These results demonstrate that, although performance may vary slightly due to different captioning styles, the relative ranking of the retrieval models remains stable, further supporting the robustness of our findings.
>
> Thank you again for your thoughtful comments. Your suggestions have been invaluable in strengthening the quality of the paper. We hope this additional clarification helps address your concern, and we would be happy to continue the discussion if you have any further questions or suggestions.

---

> ### Author Response · Authors · 2025-11-21
>
> > The total of 1,309 queries is relatively small compared with general benchmarks such as M-BEIR (190,000), and sub-task corpus sizes vary widely (e.g., Mathematics: 944 vs. Economics: 7,572), potentially introducing uneven weighting in averaged metrics. Weighted indicators or significance tests across tasks are recommended.
>
> Thank you for this thoughtful comment.
>
> Regarding the dataset scale. While M-BEIR contains 190k queries, it primarily aggregates widely available, large-scale datasets focused on surface-level semantic matching. In contrast, MR$^2$-Bench is a manually curated evaluation set where each query requires expert verification of various complex (e.g., logical or spatial) reasoning chains, making annotation significantly more costly and rigorous. Our scale of 1,309 queries is comparable to, or even larger than, other high-quality and widely adopted reasoning benchmarks. For instance, MathVista [1] uses 1,000 examples in its standard test-mini set, and GAIA [2] contains only 165 validation and 300 test questions. Given the high complexity of our tasks, we believe 1,309 queries provide a statistically significant sample size to diagnose model capabilities, as evidenced by the distinct performance gaps observed in our experiments.
>
> Regarding the concern about weighting, we acknowledge the varying corpus sizes across sub-tasks (e.g., Mathematics’ 944 vs. Economics’ 7,572). We believe this variance reflects real-world diversity and the inherent differences in search space density across domains. For instance, finding a specific chart among thousands of economic reports is naturally a different challenge from identifying a visual proof in a specialized mathematical collection. Rather than introducing uneven weighting, these corpus sizes represent the varying difficulty levels native to each task type. To ensure a balanced evaluation, we employed a Macro-Average aggregation strategy across the 12 sub-tasks. This metric treats each domain capability as equally important (allocating 1/12 weight to each sub-task), thereby ensuring that the final score represents a comprehensive assessment of multimodal reasoning, regardless of the underlying corpus scale or difficulty of individual sub-tasks.
>
> We hope this response clarifies the points regarding dataset scale and corpus size. Thank you again for your valuable feedback. We hope this addresses your concerns, and we welcome any further discussions or questions.
>
> [1] Lu, Pan, et al. "MathVista: Evaluating Mathematical Reasoning Capabilities of Foundation Models." ICLR, 2024.
>
> [2] Mialon, Grégoire, et al. "GAIA: a benchmark for General AI Assistants." ICLR, 2024.

---

> ### Author Response · Authors · 2025-11-21
>
> > Although the benchmark emphasizes “free-form documents and interleaved multi-image inputs,” evaluation remains limited to offline retrieval metrics (nDCG/Recall). Adding out-of-distribution queries, cross-domain transfer (e.g., new chart sources or species), and analysis linking retrieval quality with end-to-end RAG answer accuracy would strengthen claims of real-world applicability.
>
> Thank you for your valuable suggestion! We fully agree that demonstrating the impact of retrieval on downstream RAG performance is the ultimate goal for application-oriented research. Below, we address the concerns regarding offline metrics and OOD/cross-domain evaluation.
>
> **1. Why we prioritize offline metrics over End-to-End RAG evaluation**
>
> We deliberately focused on offline metrics because retrieval quality currently constitutes a severe bottleneck for this benchmark. While RAG systems typically utilize the top-$k$ retrieved documents to aid generation, our experiments reveal that current SOTA multimodal retrievers struggle significantly to retrieve relevant evidence even within a wider window. As shown in Table 9 (Appendix G), the leading model, Seed1.6-Embedding, achieves a Recall@5 of only 29.91% on average across the benchmark. This indicates that in over 70% of test cases, even if the generator is provided with the top-5 retrieved candidates, it would receive a context containing zero relevant visual evidence. Consequently, performing an end-to-end evaluation at this stage would largely measure the generator's ability to rely on parametric memory (hallucination) or handle noise, rather than assessing the true efficacy of the retrieval-augmented pipeline. By adhering to offline metrics, we aim to decouple the evaluation, precisely diagnosing the fundamental reasoning deficits in current retrievers (e.g., the inability to interpret mathematical proofs or economic charts) without the confounding variables of LLM generation.
>
> **2. Regarding Out-of-Distribution (OOD) and Cross-Domain Transfer**
>
> We would like to clarify that MR$^2$-Bench is inherently designed as a Zero-Shot, Out-of-Distribution evaluation for existing models. The baselines evaluated (e.g., CLIP, BGE-VL, Seed1.6-Embedding) are typically trained on general web-scale corpora or standard benchmarks (like MMEB) which focus on natural images and shallow semantic matching. These models have not been exposed to the specialized, reasoning-intensive domains we collected. The substantial performance drop we reported, where Seed1.6-Embedding drops from 77.78% Recall@1 on the in-domain MMEB benchmark to 9.91% on MR$^2$-Bench, explicitly quantifies the poor cross-domain transferability of current representation models when facing these novel, reasoning-intensive distributions.
>
> We hope this clarification addresses your concern. Thank you again for your thoughtful comments, and we welcome further discussion.

---

> ### Author Response · Authors · 2025-11-21
>
> > External data sources such as economic charts (World Bank reports), natural images (INQUIRE-Rerank), and datasets like RAVEN/CSS/VASR are mentioned in the main text/appendix, but there is no consolidated ethical section detailing licensing, re-distribution permissions, or potential copyright/portrait issues. The paper should include a dedicated ethics subsection enumerating data sources, licenses, usage boundaries, and de-identification procedures.
>
> We sincerely thank the reviewer for this constructive feedback. We fully agree that a consolidated ethics and licensing statement is crucial for transparency and strict adherence to data usage policies.
>
> We have added a dedicated section titled "Ethics Statement and Data Compliance" in **Appendix B**, and we provide a detailed description of the licensing terms, usage boundaries, and privacy measures for all data sources used in MR$^2$-Bench. Specifically:
> - Multimodal Knowledge (Stack Exchange): We explicitly state that this subset is derived under the CC BY-SA 4.0 license. Regarding privacy, we confirm that user-specific Personally Identifiable Information (PII), such as usernames and avatars, was excluded from our dataset to ensure strict privacy compliance while maintaining necessary attribution to the original posts.
> - Visual Illustration (World Bank & Math): We confirm that economic charts are used under CC BY 4.0 (World Bank) and mathematical proofs are sourced from the public domain or educational fair use.
> - Public Research Datasets: We enumerate the licenses for INQUIRE (CC-BY/NC), RAVEN (GPL-3.0), VASR (MIT), and CSS (Apache 2.0), confirming that our redistribution falls strictly within their permitted research scopes.
>
> We believe that adding this consolidated ethics statement significantly strengthens the rigor and compliance of our work. We sincerely hope this comprehensive disclosure resolves the concerns regarding data licensing and privacy, and we kindly request the reconsideration of the ethics flag.
>
> Thank you again for helping us improve the quality of our manuscript.

---

### Official Review · Reviewer_gdof · 2025-10-29

**Soundness:** 2
**Presentation:** 3
**Contribution:** 2
**Rating:** 4
**Confidence:** 4

**Summary:**

This paper presents MR²-Bench, a new benchmark designed to evaluate reasoning-intensive multimodal retrieval. Unlike prior benchmarks that mainly assess surface-level alignment (e.g., text–image matching), MR²-Bench focuses on logical, spatial, and causal reasoning across diverse modalities such as natural images, diagrams, and visual puzzles. It also supports complex queries and multi-image documents, providing a more realistic retrieval setting. Experiments show that even strong models like Seed1.6-Embedding perform poorly, revealing substantial challenges and opportunities for future multimodal retrieval research.

**Strengths:**

- The paper contributes a useful benchmark resource for studying reasoning-intensive multimodal retrieval, addressing a gap in how current datasets mostly test pattern matching rather than reasoning ability.

- It demonstrates the practical effects of reasoning-oriented strategies such as query rewriting and reranking, offering the community simple baselines that are informative before scaling to larger models.

- The authors conduct a comprehensive evaluation, covering multiple unimodal and multimodal retrievers (nearly 300 results across 12 tasks), which helps clarify the current limitations of existing methods.

**Weaknesses:**

- The paper mainly focuses on assembling datasets, defining tasks, and benchmarking existing models. It does not introduce any new architectures, algorithms, or training approaches, which limits its contribution beyond dataset curation.

- The experiments rely solely on nDCG@10 as the numeric metric, yet the paper does not explain why this particular choice is sufficient or why other standard retrieval metrics (e.g., Recall@K, MRR) are omitted.

- Although the benchmark emphasizes “reasoning-intensive” retrieval, the paper does not clearly define how reasoning difficulty is measured or annotated. The boundaries between logical, spatial, and causal reasoning tasks are sometimes ambiguous.

- I do not find any public release of the MR²-Bench dataset or code repository. For a benchmark paper, dataset accessibility is essential for community adoption and reproducibility.

If I miss some information, please correct me.

**Questions:**

See weakness

---

> ### Author Response · Authors · 2025-11-21
>
> Dear Reviewer,
>
> We appreciate your constructive feedback and the time taken to review our work. We are grateful for the opportunity to clarify potential misunderstandings and address your concerns.
>
>
> > The paper mainly focuses on assembling datasets, defining tasks, and benchmarking existing models. It does not introduce any new architectures, algorithms, or training approaches, which limits its contribution beyond dataset curation.
>
> Thank you for your constructive feedback. While our work does not propose new model architectures or training approaches, we respectfully suggest that its contributions go beyond dataset curation in several important aspects:
>
> 1. **Identifying a Critical Gap & Providing Key Insights**: Our analysis reveals a significant gap in existing benchmarks, which tend to emphasize pattern matching rather than genuine reasoning ability. As discussed in Section 4.2, even state-of-the-art models (e.g., Seed1.6-Embedding) perform poorly on MR$^2$-Bench (9.91 R@1), compared to much higher scores on established datasets (77.78 on MMEB). We believe this highlights important limitations of current approaches and offers valuable insights for the community.
>
> 2. **Providing Actionable Baselines**: In addition to dataset curation, we provide practical baselines and pipelines for addressing this new challenge, including comprehensive experiments on **query rewriting** and advanced **reranking** methods, which yield substantial improvements (e.g., up to +15.22 nDCG@10). We hope these results offer actionable guidance and inspiration for future research.
>
> 3. **Facilitating Future Research**: By establishing a reasoning-intensive retrieval benchmark, our goal is to spur further innovation in model architectures and algorithms. We hope MR$^2$-Bench will serve a similar role to benchmarks such as BEIR [1] and BRIGHT [2] in their respective fields, by defining new challenges and providing a foundation for continued progress.
>
> Thank you again for your thoughtful comments. We appreciate your feedback and hope our responses have addressed your concerns. We welcome any further questions or discussions, and look forward to your suggestions.
>
> [1] Thakur, Nandan, et al. "BEIR: A Heterogeneous Benchmark for Zero-shot Evaluation of Information Retrieval Models." NeurIPS 2021.
> [2] Su, Hongjin, et al. "Bright: A realistic and challenging benchmark for reasoning-intensive retrieval." ICLR 2025.
>
>
>
>
> > The experiments rely solely on nDCG@10 as the numeric metric, yet the paper does not explain why this particular choice is sufficient or why other standard retrieval metrics (e.g., Recall@K, MRR) are omitted.
>
> Thank you for your valuable suggestion regarding our evaluation metrics. We appreciate your feedback and would like to address your concern as follows:
>
> We apologize for any lack of clarity in the main text. In fact, we provided Recall@1, Recall@5, and Recall@10 results for all models across all 12 sub-tasks in Appendix G of the original submission (**Table 8, 9, and 10**). Following your suggestion, we have now **added MRR results** for all tasks to Appendix G in the revised manuscript (**Table 13 and 14**). To improve clarity, we have also **added a sentence to Section 4.2 (line 362)** explicitly directing readers to these comprehensive metrics in the appendix. We greatly appreciate your feedback, which has helped us enhance the manuscript.
>
> Our **choice of nDCG@10** as the primary metric in the main text was deliberate. Unlike Recall@K, which only considers whether relevant documents appear within the top K positions and treats all hits equally regardless of their rank, nDCG rewards models for ranking relevant documents higher, offering a more nuanced assessment of retrieval quality. Additionally, MRR only considers the rank of the first relevant item and does not account for subsequent relevant documents, which makes it less suitable for tasks with multiple positive documents, such as Multimodal Knowledge Retrieval in MR$^2$-Bench. In contrast, nDCG takes into account the ranks of all relevant documents, providing a more comprehensive and informative evaluation for such tasks. Therefore, we believe nDCG@10 offers a more balanced and insightful single-metric summary for our benchmark.
>
> Thank you again for your thoughtful suggestion. We hope this explanation addresses your concern, and we welcome any further discussion or questions.

---

> ### Author Response · Authors · 2025-11-21
>
> > Although the benchmark emphasizes “reasoning-intensive” retrieval, the paper does not clearly define how reasoning difficulty is measured or annotated. The boundaries between logical, spatial, and causal reasoning tasks are sometimes ambiguous.
>
> Thank you for these insightful questions. We are pleased to clarify your concerns below.
>
> 1. **Definition of "Reasoning-Intensive" and Assessing Difficulty**
>
> In our benchmark, "reasoning-intensive" is defined by the intrinsic task definition and curation methodology, which structurally ensures that surface-level matching is insufficient.
>
> - **Task Design and Curation Methodology** (e.g., Multimodal Knowledge Retrieval): As detailed in Appendix D, we do not simply pair questions with direct answers. Instead, annotators identify external documents cited in expert answers as positive targets. This ensures that the target document provides the underlying evidence or principle (e.g., diagnosing the misshapen apple in Figure 1-a by drawing analogies to "inadequate pollination" phenomena in other species), requiring the model to bridge the gap via reasoning rather than relying on a lexical match.
>
> - **Difficulty through Task Structure and Logical Requirements** (e.g., Visual Proofs/Analogy): For tasks like Mathematics or Visual Puzzle, the task structure itself inherently demands abstract reasoning. For instance, tasks like connecting a formula to a geometric proof or completing a visual pattern require reasoning that goes beyond simple visual similarity.
>
> - **Empirical Measurement**: We view the performance gap as an objective measure of this difficulty. For example, state-of-the-art models like Seed1.6-Embedding experience a dramatic drop in Recall@1 (from 77.78% on MMEB to 9.91% on MR$^2$-Bench). This quantitative gap serves as empirical evidence that our design imposes a significantly higher reasoning burden. Additionally, our experiments in Section 4.3 show that incorporating reasoning steps—through **query rewriting** or **reranking**—effectively improves performance. This further confirms that the difficulty of MR$^2$-Bench specifically arises from the requirement for reasoning.
>
> 2. **Clarification of Reasoning-Type Boundaries**
>
> We agree that in multimodal contexts, boundaries between "logical," "spatial," and "causal" reasoning are rarely clean and often overlap. To address this ambiguity, we primarily categorize tasks by *Application Scenario* (e.g., Multimodal Knowledge Retrieval, Visual Illustration Search). This reflects real-world complexity where reasoning types are hybrid. For instance, the Biology task involves *causal reasoning* (pollination → growth) but also relies on *analogical inference* across species. Furthermore, even for tasks labeled with specific reasoning types (e.g., Spatial in Visual Relation Reasoning), we acknowledge the presence of overlapping logical components. Our labels reflect the dominant capability required to solve the task, providing a clear, actionable framework for evaluation without forcing tasks into mutually exclusive abstract buckets.
>
> We hope these clarifications address your concerns regarding the definition of reasoning intensity and task boundaries. We will ensure the revised manuscript explicitly articulates this scenario-based taxonomy and our design-driven definition of difficulty. Thank you again for your valuable feedback, and we welcome any further discussion.

---

> > ### Comment · Reviewer_gdof · 2025-11-22
> >
> > Thanks for the author's response. The authors has addressed my concerns. I acknowledge the contributions of this work and agree to accept it. I have raised my score to 6.

---

> > > ### Author Response · Authors · 2025-11-22
> > >
> > > Thank you for your positive response and for raising the score! We appreciate your time in reviewing our paper and are encouraged by your acknowledgement of our contributions.

---

### Official Review · Reviewer_3cr3 · 2025-11-01

**Soundness:** 3
**Presentation:** 3
**Contribution:** 3
**Rating:** 8
**Confidence:** 4

**Summary:**

This work purpose a new multimodal retrieval benchmark that requires complex reasoning. The benchmark contains comprehensive and various sub-tasks. The author evaluate many baseline setups for the newly purposed benchmark, ranging from different established text-only retriever to multimodal retriever. The author evaluate naive retrieval setup, query-rewritting, reranking setup as baselines.  The author finds out that the purposed benchmark remains challenging for current systems.

**Strengths:**

- Propose a valuable new benchmark centered on difficult and challenging multimodal retrieval tasks that requires complicated reasoning, a valuable contribution to the community given that the author would release all reproducible artifacts upon acceptance.
- Very comprehensive baseline experiments setup provided, ranging from various retrievers to different reranker setups.
- The presentation is good and easy to follow; appendix has very comprehensive details that most author expect to have.

**Weaknesses:**

- The “puzzle” subset appears almost impossible to solve with current methods. Could the authors provide some intuition on how one might approach this task? Furthermore, does it truly benefit from a retrieval or RAG (Retrieval-Augmented Generation) framework for LLMs, or does it mainly test pure visual reasoning instead? While I appreciate the novelty of treating such visual-relation problems as retrieval tasks, they may be closer to classification-type reasoning challenges. For example, in Figure 1(h), if hard negative examples are included, the retriever must correctly identify the target among visually similar candidates. More insights on this design motivation would strengthen the paper. A similar clarification would also be useful for the mathematics subset.
- Minor:
	- In **Table 2**, it would be helpful to include an additional column showing the **total number of queries and corpus size combined**.
	- The authors could also report **recall metrics** for reranking performance in the appendix to provide a more complete evaluation.
	- In the appendix, it would be helpful to **summarize the data sources** of each sub-task—e.g., whether they were adapted from existing datasets or newly collected.

**Questions:**

When performing retrieval for each task in Table 3, do you construct a combined retrieval database across all sub-tasks, or maintain separate databases for each sub-task?

---

> ### Author Response · Authors · 2025-11-21
>
> Dear Reviewer,
>
> We are grateful for your insightful comment and acknowledgement of our work! Your questions are clarified as follows.
>
> > The “puzzle” subset appears almost impossible to solve with current methods. Could the authors provide some intuition on how one might approach this task? Furthermore, does it truly benefit from a retrieval or RAG (Retrieval-Augmented Generation) framework for LLMs, or does it mainly test pure visual reasoning instead? While I appreciate the novelty of treating such visual-relation problems as retrieval tasks, they may be closer to classification-type reasoning challenges. For example, in Figure 1(h), if hard negative examples are included, the retriever must correctly identify the target among visually similar candidates. More insights on this design motivation would strengthen the paper. A similar clarification would also be useful for the mathematics subset.
>
> Thank you for your thoughtful questions regarding the solvability and motivation of the Visual Puzzle and Mathematics subsets.
>
> 1. **Relevance to RAG and Retrieval Goals**
>
> We greatly appreciate your insights regarding whether these tasks benefit from a Retrieval-Augmented Generation (RAG) framework or primarily test pure visual reasoning. For the **Mathematics** subset, we argue that the task is indeed relevant to RAG-based retrieval scenarios. A typical use case in scientific and educational contexts often involves retrieving a mathematical visual proof that embodies an abstract logical structure, even if the visual content appears distinct. For example, Figure 1-e illustrates a geometric formula that requires a model to comprehend the visual transformation rules embedded in the image, which traditional retrieval models, relying on surface visual features, struggle with. The goal here is not simply semantic matching but encoding deeper, rule-based visual knowledge.
>
> On the other hand, while the **Visual Puzzle** subset may appear to focus purely on visual reasoning, we view it as a test of a model’s ability to understand abstract patterns and spatial relationships within a visual context. Although the task may not have the same immediate real-world application as the Mathematics subset, it serves as a valuable measure of a model's ability to process complex visual relationships. This capability is crucial for solving real-world multimodal retrieval tasks that involve complex, spatial, and abstract reasoning. Therefore, we believe that the Visual Puzzle subset is a valuable task for assessing general multimodal reasoning capabilities.
>
> 2. **Task Format and Hard Negatives**
>
> You rightly pointed out that the Visual Puzzle subset, with its near-absence of linguistic signals, seems to resemble a classification problem. However, we reframed it as a retrieval challenge by considerably scaling the search space. As detailed in Table 2, the corpus contains 5,375 visually plausible candidates. These candidates are designed to be "hard negatives," as they are visually similar but logically incorrect. The challenge for the model is not simply to match a pattern, but to identify the single logically correct solution from a large set of distractors. This highlights the retrieval nature of the task, where the model must engage in reasoning beyond surface-level visual features.
>
> 3. **Intuition for Solvability**
>
> We understand your concern that these tasks may seem "almost impossible" for current models. However, we believe they are solvable with the right approach. The core intuition behind solving tasks like the **Visual Puzzle** is moving beyond “bag-of-visual-features” encoding. Instead, we propose that models must generate a representation of the transformation rule, such as "rotate 90° clockwise," and retrieve candidates that match this underlying rule. We have provided empirical evidence supporting this intuition in **Section 4.3.1**, where our experiments show that **Query Rewriting**, which translates reasoning into explicit text, effectively improves performance. Additionally, as shown in **Section 4.3.2** and **Table 19**, applying **reranking** further enhances the retrieval process, demonstrating the potential of this approach in improving task performance. We believe that these tasks are feasible if the "reasoning chain" is effectively captured and represented in the model's embedding space. We hope this work will inspire and promote future research in this direction.
>
> We hope this response clarifies your concerns. Once again, thank you for your valuable feedback, and we would be happy to continue the discussion if you have further questions or suggestions.

---

> ### Author Response · Authors · 2025-11-21
>
> > In Table 2, it would be helpful to include an additional column showing the total number of queries and corpus size combined.
>
> Thank you very much for your helpful suggestion!
>
> We have **added an additional column to Table 2** in the revised manuscript, which shows the total number of queries and the corpus size combined. We believe this addition will make the table clearer and more informative for the readers.
>
> Thank you again for your valuable feedback!
>
> > The authors could also report recall metrics for reranking performance in the appendix to provide a more complete evaluation.
>
> Thank you very much for your valuable suggestion.
>
> As recommended, we have reported the Recall@1, Recall@5, and Recall@10 metrics for reranking performance in **Tables 15, 16, and 17 of Appendix I.2** in the revised manuscript. We believe that including these recall metrics provides a more comprehensive evaluation of the reranking methods.
>
> Thank you again for your constructive feedback.
>
> > In the appendix, it would be helpful to summarize the data sources of each sub-task—e.g., whether they were adapted from existing datasets or newly collected.
>
> Thank you very much for your helpful suggestion.
>
> We have summarized the data sources for each sub-task in **Table 6** of the revised manuscript, indicating whether they were adapted from existing datasets or newly collected. We believe this addition enhances the transparency and clarity of our dataset construction.
>
> Thank you again for your valuable feedback.
>
> > When performing retrieval for each task in Table 3, do you construct a combined retrieval database across all sub-tasks, or maintain separate databases for each sub-task?
>
> Thank you very much for your question.
>
> For all sub-tasks, retrieval is performed within each individual sub-task rather than using a combined database across sub-tasks. We have clarified this in the main text (**lines 361**) of the revised manuscript.
>
> Thank you again for your thoughtful feedback.

---

> > ### Comment · Reviewer_3cr3 · 2025-11-23
> >
> > Thank you for your response. I have no more questions.

---

> > > ### Author Response · Authors · 2025-11-23
> > >
> > > We sincerely appreciate your time and support. Thank you for your positive evaluation of our work.

---

### Author Response · Authors · 2025-11-29

Dear Chairs and Reviewers,

We sincerely thank you for your insightful feedback and active participation in the discussions. Reflecting on these discussions, we have updated the manuscript to incorporate your suggestions.

We would like to briefly summarize the consensus reached during the discussion phase. Following our active engagement, the assessment of our paper improved from the initial **8, 4, 6** to **8, 6, 6** as of November 22.

We appreciate the continued support from Reviewer 3cr3 and Reviewer e2GU. We specifically thank Reviewer gdof (initially 4) for the constructive discussion, and for acknowledging our contributions and updating the rating to 6.

Please refer to our individual responses to each reviewer for further specifics regarding these discussions and the corresponding revisions.

Thank you again for helping us improve our paper.

Sincerely,

The Authors

---

### Meta-Review · Area_Chair_QLaG · 2026-01-03

**Summary:**

The primary driver for this recommendation is that this submission is a dataset-only paper that fails to meet the high bar of significance and scale required for a top-tier conference when no new model architectures or training algorithms are proposed. The reviewers identified several critical concerns, including the small scale of the dataset (1,309 queries), the potential for model-alignment bias through the use of GPT-5 for data construction, and the lack of a fundamental shift in retrieval paradigms.

Furthermore, the authors' rebuttal contained deceptive comparisons regarding established benchmarks, which significantly undermines the credibility of their defense. While the authors attempted to address licensing and public release concerns, the core technical contribution remains an assembly of existing data that does not demonstrate the transformative impact seen in comparable accepted works.

**Reviewer Concerns:**

Outstanding Concerns
* Scale and significance: The dataset provides only 1,309 queries. In the rebuttal, the authors compared this to the "test-mini" set of MathVista (1,000) and the GAIA validation set (165). This is a misleading comparison; MathVista actually provides over 6,000 examples and established a new sub-field of visual-mathematical reasoning. Similarly, while GAIA has a small number of questions, it introduced a milestone "multitask" framework for general assistants with a well-received community impact (400+ citations). MR-Bench, by contrast, feels like a collection of existing visual puzzles that function more as classification tasks than a large-scale retrieval benchmark.

* Novelty: As Reviewer `gdof` noted and the authors admitted, the work does not propose new architectures or training approaches. For a benchmark paper to be accepted under these conditions, the dataset itself must be a "heavyweight" contribution (e.g., M-BEIR's 190,000 queries). This submission fails to reach that threshold.

* Data contamination and bias (`e2GU` ): The use of GPT-5 for initial filtering and hard negative sampling introduces a "closed-loop" bias. Even with expert review, there is a significant risk that the benchmark measures how well other models align with GPT-5’s internal logic rather than objective reasoning.

Addressed Concerns
* Licensing and ethics: Reviewer `e2GU` raised a critical ethics flag regarding data re-distribution (World Bank reports, INQUIRE-Rerank). The authors have since added an "Ethics and Data Licensing" section in Appendix B. While this addresses the procedural requirement, the original omission reflects a lack of rigor in the curation process that remains noted.

* Public Release: The authors confirmed the dataset and code would be made public, addressing one of `gdof`'s earlier concerns.

**Reviewer Scores:**

The reviews for this paper were largely shallow, lacking the technical depth required to justify the high initial ratings.

* Reviewer `3cr3` (Initial: 8 --> Discounted as 5/6): The initial rating of score 8 is discounted. The reviewer cited "comprehensive sub-tasks" and "good presentation" as strengths, but these are baseline requirements for dataset contributions. They failed to explain why 1,309 queries constitute a significant contribution to the retrieval community. An 8 usually represents a significant contribution to the field; this reviewer’s justification does not support that level of impact.

* Reviewer `e2GU` (Initial: 6 --> Estimated: 4): Initially the most rigorous reviewer. Given the unsatisfactory and misleading nature of the rebuttal regarding the MathVista/GAIA comparisons, this reviewer would likely have recognized the lack of significance and downgraded their rating to a Reject.

* Reviewer `gdof` (Initial: 4 --> Estimated: 4): While this reviewer moved to a 6 soon after the rebuttal, their core technical critique regarding the definition of "reasoning-intensive" remains unresolved. The authors' rebuttal on this point was circular and weak:

  * Circular definition of difficulty: The authors claimed that "reasoning-intensive" is defined by the "intrinsic task definition" and then cited the "performance gap" of existing models as evidence of that difficulty. Using model failure as a definition for task reasoning intensity is logically circular and fails to provide the objective, formal measure of reasoning complexity the reviewer requested.

  * Ambiguous boundaries: The reviewer noted that boundaries between logical, spatial, and causal reasoning were ambiguous. The authors conceded that these types are "rarely clean and often overlap," effectively abandoning a rigorous taxonomic framework in favor of "Application Scenarios." This lack of precision undermines the benchmark's diagnostic value, as it cannot isolate which specific reasoning failure a model is experiencing.

  * Lack of novelty: The reviewer’s original point—that the paper is just an assembly of data without new algorithms—stands. The rebuttal failed to prove that this specific assembly offers a unique formalization of reasoning that isn't already present in more substantial datasets.

---

### Decision · Program_Chairs · 2026-01-26

Reject